



# The substructure of extremely hot summers in the Northern Hemisphere

Matthias Röthlisberger[1], Michael Sprenger[1], Emmanouil Flaounas[1], Urs Beyerle[1] and Heini Wernli[1]

[1]Institute for Atmospheric and Climate Science, ETH Zürich, Zürich, Switzerland

*Correspondence to*: Matthias Röthlisberger (matthias.roethlisberger@env.ethz.ch)

**Abstract.** In the last decades, extremely hot summers (hereafter extreme summers) have challenged societies worldwide through their adverse ecological, economic and public health effects. In this study, extreme summers are identified at all grid points in the Northern Hemisphere in the upper tail of the July–August (JJA) seasonal mean 2-meter temperature (T2m) distribution, separately in ERA-Interim reanalyses and in 700 simulated years with the Community Earth System Model (CESM) large ensemble for present-day climate conditions. A novel approach is introduced to characterize the substructure of extreme summers, i.e., to elucidate whether an extreme summer is mainly the result of the warmest days being anomalously hot, or of the coldest days being anomalously mild, or of a general shift towards warmer temperatures on all days of the season. Such a statistical characterization can be obtained from considering so-called rank day anomalies for each extreme summer, that is, by sorting the 92 daily mean T2m values of an extreme summer and by calculating, for every rank, the deviation from the climatological mean rank value of T2m.

Applying this method in the entire Northern Hemisphere reveals spatially strongly varying extreme summer substructures, which agree remarkably well in the reanalysis and climate model data sets. For example, in eastern India the hottest 30 days of an extreme summer contribute more than 70% to the total extreme summer T2m anomaly, while the colder days are close to climatology. In the high Arctic, however, extreme summers occur when the coldest 30 days are substantially warmer than climatology. Furthermore, in roughly half of the Northern Hemisphere land area, the coldest third of summer days contribute more to extreme summers than the hottest third, which highlights that milder than normal coldest summer days are a key ingredient of many extreme summers. In certain regions, e.g., over western Europe and western Russia, the substructure of different extreme summers shows large variability and no common characteristic substructure emerges. Furthermore, we show that the typical extreme summer substructure in a certain region is directly related to the region's overall T2m rank day variability pattern. This indicates that in regions where the warmest summer days vary particularly strongly from one year to the other, these warmest days are also particularly anomalous in extreme summers (and analogously for regions where variability is largest for the coldest days). Finally, for three selected regions, thermodynamic and dynamical causes of extreme summer substructures are briefly discussed, indicating that, for instance, the onset of monsoons, physical boundaries like the sea ice edge, or the frequency of occurrence of Rossby wave breaking, strongly determine the substructure of extreme summers in certain regions.



# 1 Introduction

During the last decades, numerous high-impact hot temperature extremes occurred on approximately seasonal time scales,
including the extremely hot European summer in 2003 (Fink et al., 2004; Schär and Jendritzky, 2004), the 2010 Russian heat
wave (Barriopedro et al., 2011), the hot and dry summer 2015 in Europe (Dong et al., 2016; Hoy et al., 2017; Orth et al., 2016),
the hot and humid summer 2015 in western India and Pakistan (Wehner et al., 2016), and the concurrent heat waves across the
Northern Hemisphere in the summer 2018 (Vogel et al., 2019). It is well known that individual heat waves on time scales of
up to a few weeks cause societal challenges, for example serious public health issues (e.g., Fouillet et al., 2006). However, the
large socio-economic and ecological impacts of the seasonal events listed above (e.g., Ciais et al., 2005; Buras et al., 2019)
illustrated that many economic sectors such as agriculture, tourism and re-insurance are particularly susceptible to temperature
extremes on seasonal (as opposed to synoptic) time scales. Therefore, understanding the statistical properties of entire
extremely hot summers (hereafter referred to as "extreme summers") as well as their physical causes is a research topic of high
societal relevance.


The concept of an extreme summer [as a particular type of an "extreme season", cf. Wernli et al. (in prep.)] is closely related
to the concept of a heat wave, even though there are important differences. An individual heat wave is commonly understood
to be a single, quasi-continuous episode of abnormally hot surface weather with a duration ranging from days to weeks (Russo
et al., 2015; Zschenderlein et al., 2019). Heat waves are thus strongly influenced by individual synoptic flow features such as
atmospheric blocks (Brunner et al., 2017; Pfahl and Wernli, 2012; Röthlisberger and Martius, 2019; Zschenderlein et al., 2019),
stationary ridges (Sousa et al., 2018) or recurrent Rossby wave patterns (Röthlisberger et al., 2019). In contrast, extreme
summers have a fixed duration (of three months), which is beyond the time scale of these synoptic flow features. Consequently,
extreme summers require a temporal organization of the relevant synoptic flow features, which can occur either "by chance"
(internal atmospheric variability) or favored by more slowly varying processes. Possible candidates for the latter are soil
moisture fluctuations (Fischer et al., 2007; Lorenz et al., 2010; Seneviratne et al., 2010), sea ice dynamics (Cohen et al., 2014)
or large-scale modes of variability in the ocean and atmosphere (e.g., Schneidereit et al., 2012). Understanding how this
temporal organization of weather within seasons occurs is challenging as it requires a seamless approach (Hoskins, 2013),
which couples weather system dynamics to these more slower varying processes.

In contrast to individual heat waves, extreme summers may be composed of one or several hot periods. In some cases, an
extreme summer may even contain rather cool periods separating individual heat waves. Thus, extreme summers exhibit
distinct "substructures", some of which are schematically illustrated in Fig. 1. For example, a summer might be an extreme
summer because the hottest days of the season are particularly anomalous, with the remainder of the summer days being only
moderately warmer than or even close to climatology. Such an extreme summer substructure was observed in large parts of
Europe in the summer 2015, when the anomalies of the seasonal hottest days exceeded those of the seasonal mean by almost



a factor of two (Dong et al., 2016). Hence the hottest days of the 2015 summer contributed over proportionally to the seasonal mean anomaly. However, also other substructures are plausible: several episodes of hot but not extreme temperatures, perhaps even interrupted by fewer cool episodes, a suppression of cool summer days, or a uniform shift in the entire summer temperature distribution.


Knowledge about the extreme summer substructure is relevant for at least two reasons. Firstly, the societal impact of an extreme summer featuring one (or several) periods of extremely hot temperatures (i.e., hottest summer days being hotter than normally) will likely differ from the societal impact of an extreme summer resulting primarily from a suppression of cool summer days (i.e., coldest summer days being milder than normally), or from an extreme summer characterized by a uniform shift in the

entire temperature distribution (i.e., all summer days warmer than normally). Secondly, also the physical and meteorological causes of extreme summers with such distinct substructures conceivably differ. Thus, identifying the substructure of extreme summers is likely a starting point for understanding also their physical causes.

The purpose of this study is to characterize extreme summers statistically by quantifying their substructure. To do so, we define

extreme summers in the upper tail of the June–August (JJA) mean two-meter temperature (T2m) distribution. Thereafter, the extreme summer substructure is assessed by decomposing the seasonal mean T2m anomaly of a particular extreme summer into the contributions from all rank days of that season (i.e., the contribution from the coldest day, the second coldest day etc.). This decomposition thus allows to quantify the contributions from all parts of the T2m distribution (e.g., the coldest, middle and hottest thirds of summer days) to the seasonal T2m anomaly of an extreme summer.


Here we use the ERA-Interim re-analysis data set to study the substructure of past extreme summers. However, extreme summers are by definition extremely rare events. Thus, in order to yield robust results, a climatological investigation of the extreme summer substructure requires much longer data records than provided by ERA-Interim or any other currently available high-quality re-analysis data set. We therefore complement ERA-Interim with a 700-year present day climate simulation (for

details, see Sect. 2.2) to address the following research goals:

1. Propose and illustrate a simple method for decomposing the seasonal mean temperature anomaly into its contributions from each rank day.

2. Use this decomposition to analyze the substructure of extreme summers separately at selected grid points.

3. Quantify and compare the spatial variability in extreme summer substructures in the Northern Hemisphere in both re-
analysis and climate model data.

4. Illustrate physical causes of the observed (and simulated) extreme summer substructures in selected regions.



## 2 Data and Methods

### 2.1 ERA-Interim

We use ERA-Interim re-analysis data (Dee et al., 2011) covering the period 1979-2018. ERA-Interim is originally produced
with a T255 spectral horizontal resolution and 60 hybrid σ-$p$ levels in the vertical. We interpolated the data horizontally to a
1° by 1° grid and vertically to pressure and isentropic levels. The ERA-Interim data is provided at 6-hourly time intervals, in
this study however, we aggregated all data to a daily temporal resolution. Besides the T2m fields, we also use potential vorticity
(PV), total precipitation, 250 hPa meridional winds and sea ice concentration. Furthermore, we deliberately do not detrend the
ERA-Interim T2m data as we are interested in extreme summers exhibiting the largest absolute T2m anomalies and not the
largest T2m anomalies relative to a long-term trend.

### 2.2 CESM

Besides ERA-Interim, the Community Earth System Model version 1 (CESM, Hurrell et al., 2013) is used to perform present-
day climate simulations using restart files from the CESM large ensemble project (CESM-LENS, Kay et al., 2015). We use
atmospheric fields at daily temporal resolution, with a horizontal resolution of approximately 1° and 30 vertical levels. The
original CESM-LENS data contains a 35-member ensemble of simulations started on 1 January 1920 and integrated forward
in time until 2100. These 35 "macro ensemble" members were rerun for the period from 1 January 1990 to 31 December 1999
in order to obtain temporally high-resolution three-dimensional model output. To further increase the number of simulated JJA
seasons, a "micro ensemble" with additional 35 members was branched off from member one of the macro ensemble, on 1
January 1980, by adding an $\mathcal{O}(10^{-13})$ perturbation to the initial atmospheric temperature field of each micro ensemble. These
additional micro ensemble runs are then integrated forward in time until 31 December 1999. Fischer et al. (2013) have shown
that at the latest after a decade, the micro ensemble members exhibit a similar spread in atmospheric variables compared to
members of the macro ensemble. Thus, for the period 1990–1999, the micro ensemble members can be regarded as additional
independent members, yielding a total of 70 ensemble members covering the 10-year period from 1990–1999, i.e., 700 years
of present-day climate.

### 2.3 Decomposing a seasonal T2m anomaly to quantify the season's substructure

To examine the substructure of a particular July–August (JJA) season $k$, we decompose its seasonal T2m anomaly ($SA_k$) into
contributions from the ranked $D$ daily T2m values of season $k$, where $D$ is the number of days in season $k$ (e.g., for JJA $D = 92$). We thus aim to quantify how much each rank day (i.e., coldest day, second coldest day, etc.) of season $k$ contributes to
the seasonal anomaly $SA_k$. This decomposition of $SA_k$ is illustrated for the example grid point 9°E/47°N (near Zürich,
Switzerland) in Fig. 2 and introduced more formally below. It is applied to both data sets separately in exactly the same fashion
and therefore, a superscript $M \in \{ERAI, CESM\}$ will only be used where it is necessary to explicitly distinguish between the



two datasets. All the important statistical quantities used in this study are summarized in Tab. 1. Furthermore, bear in mind that all these quantities are calculated at each grid point individually.

We start by ranking all daily mean T2m values within their respective season $k$ (Figs. 2a,b) and compute seasonal means $(SM_k)$, i.e.,

$$SM_k = \frac{1}{D}\sum_{d=1}^{D} T_{d,k}, k = 1, \ldots, K, \tag{1}$$

where $T_{d,k}$ is the daily mean T2m value with rank $d$ in season $k$ (see Fig. 2b). At each grid point we thus compute $K^{ERAI} = 40$ seasonal mean values for ERA-Interim and $K^{CESM} = 700$ values for CESM.

The climatological seasonal mean $(C)$ is also calculated from the ranked daily mean T2m values $(T_{d,k})$ as

$$C = \frac{1}{K \cdot D}\sum_{k=1}^{K}\sum_{d=1}^{D} T_{d,k} = \frac{1}{D}\sum_{d=1}^{D}\frac{1}{K}\sum_{k=1}^{K} T_{d,k}. \tag{2}$$

Hereby, $\frac{1}{K}\sum_{k=1}^{K} T_{d,k}$ is the average T2m value of all $K$ days with rank $d$ in their respective season, e.g., for $d = 1$ the average coldest day of the season and for $d = 92$ the average hottest day of the season. Hence, $C$ is computed as the mean over the average T2m values for each rank. These rank day T2m means (bold gray contour in Fig. 2b) are hereafter referred to as

$$RDM_d = \frac{1}{K}\sum_{k=1}^{K} T_{d,k}, d = 1, \ldots, D. \tag{3}$$

Using the $RDM_d$, the seasonal T2m anomaly of any season $k$ $(SA_k)$ can be decomposed into contributions from each of the $D$
rank days:

$$SA_k = SM_k - C = \frac{1}{D}\left(\sum_{d=1}^{D} T_{d,k} - \sum_{d=1}^{D} RDM_d\right) = \frac{1}{D}\sum_{d=1}^{D}(T_{d,k} - RDM_d) = \frac{1}{D}\sum_{d=1}^{D} RDA_{d,k}, \tag{4}$$

where in the last equality the rank day anomaly of the day with rank $d$ in season $k$ is introduced as $RDA_{d,k} = T_{d,k} - RDM_d$. In other words, the seasonal mean anomaly $SA_k$ is expressed as the average rank day anomaly (see also Fig. 2c).

This decomposition of $SA_k$ thus allows to assess the exact contribution from each (ranked) day of season $k$ to $SA_k$. In the
following we split the 92 days of each JJA season $k$ into three parts according to their rank and focus on the relative contributions to $SA_k$ from the coldest, middle and hottest third of the 92 days of season $k$ by calculating, e.g.,

$$SF_{cold,k} = \left(\frac{1}{D}\sum_{d=1}^{\left[\frac{D}{3}\right]} RDA_{d,k}\right)\Big/ SA_k. \tag{5}$$




The notation $[x]$ hereby stands for $x$ rounded to the nearest integer. For computing contributions to $SA_k$ from the middle and hottest thirds of the summer days ($SF_{middle,k}$ and $SF_{hot,k}$), the sum in Eq. (5) runs from $\left[\frac{D}{3}\right] + 1$ to $\left[D\frac{2}{3}\right]$ for $SF_{middle,k}$ and from $\left[D\frac{2}{3}\right] + 1$ to $D$ for $SF_{hot,k}$. By construction, the sum of the three fractions amounts to 1.


**2.4 Identification and substructure of extreme summers**

Extremely hot summers at each grid point in the Northern Hemisphere are identified in the ERA-Interim (CESM) data set as the 5 (35) hottest JJA seasons, yielding two sets of extreme summers $\mathbb{X}^M = \{k_1, \dots, k_{N^M}\}, M \in \{ERAI, CESM\}$ with $N^{ERAI} = 5$ and $N^{CESM} = 35$ members, respectively. Hence, ERA-Interim extreme summers correspond to the 12.5% hottest summers (5 out of 40), while the CESM extreme summers correspond to the 5% hottest summers (35 out of 700).

An analogous procedure to that described in Sect. 2.3 is employed to quantify the contributions from each of the three thirds of the extreme summer days to the average T2m anomaly of the $N$ considered extreme summers. The mean of these extreme summers ($XM$) is calculated as $XM = \frac{1}{N}\sum_{k \in \mathbb{X}} SM_k$ and is used to compute the mean anomaly of these extreme summers $XA = XM - C$. The relative contributions from the three thirds of the summer days to the extreme summer anomaly $XA$ are calculated as, e.g.,

$$XF_{cold} = \left(\frac{1}{N}\sum_{k \in \mathbb{X}}\frac{1}{D}\sum_{d=1}^{\left[\frac{D}{3}\right]} RDA_{d,k}\right) \Big/ XA. \tag{6}$$

The quantities $XF_{cold}$, $XF_{middle}$ and $XF_{hot}$ again add up to 1 and quantify the relative contributions from the three thirds to the average T2m anomaly of all extreme summers at a particular grid point.

**3   Results and discussion**

**3.1 Extreme summer T2m anomalies**

Figures 3a and 4a depict the average T2m anomalies during extreme summers in the two data sets ($XA^{ERAI}$ and $XA^{CESM}$, respectively). In both data sets, $XA$ exhibits considerable spatial variability. The ERA-Interim extreme summers have temperature anomalies of up to 3 K over western Russia, while over some tropical ocean areas $XA^{ERAI}$ is less than 0.5 K (Fig. 3a). The $XA^{CESM}$ field exhibits a generally similar spatial pattern to $XA^{ERAI}$, with larger values over land than over the oceans (Fig. 4a). However, $XA^{CESM}$ generally exceeds $XA^{ERAI}$, as the summers $\mathbb{X}^{CESM}$ are statistically more extreme than the summers $\mathbb{X}^{ERAI}$. In the following, we decompose the extreme summer T2m anomalies ($XA$) shown in Figs. 3a and 4a using the methodology described in Sect. 2.3 and 2.4, first at few selected grid points and then for all Northern Hemisphere grid points.





### 3.2 Extreme summer substructures at selected grid points

The rank day anomalies ($RDA_{d,k}^{ERAI}$) for the five ERA-Interim extreme summers at a grid point located in eastern India (81°E/21°N, Figs. 3a,b) reveal a similar substructure in each of the extreme summers. The largest $RDA_{d,k}^{ERAI}$ (up to 5 K) occur in the hottest 30 days of each season, while for the 60 coldest summer days in each extreme summer, $RDA_{d,k}^{ERAI}$ does not exceed 1.5 K. The contributions of the coldest, middle and hottest third of all extreme summer days to $XA^{ERAI}$ at this grid point (i.e., $XF_{cold}^{ERAI}$, $XF_{middle}^{ERAI}$ and $XF_{hot}^{ERAI}$) are 10%, 17% and 73%, respectively. For the 2005 summer, the contributions were 1%, 7%

and 92%, and hence, almost the entire seasonal T2m anomaly resulted from the hottest 30 days of the summer being hotter than normally.

A comparison between the ERA-Interim and CESM extreme summer substructures at this grid point (Figs. 3b and 4b) reveals remarkable qualitative similarities between the extreme summer substructure at 81°E/21°N in the two data sets. At this grid

point, also the season $\mathbb{X}^{CESM}$ exhibit largest $RDA_{d,k}^{CESM}$ values for the 30 hottest summer days. Moreover, despite the different number of seasons in the two data sets, the $XF_{cold}^{CESM}$, $XF_{middle}^{CESM}$ and $XF_{hot}^{CESM}$ values of 12%, 23% and 65%, respectively, are not far off the respective values for the seasons $\mathbb{X}^{ERAI}$. Figures 3b and 4b further reveal that the largest $RDA_{d,k}^{CESM}$ values reach much larger values (up to 8 K) than the $RDA_{d,k}^{ERAI}$ values, which is an expected result, since the seasons $\mathbb{X}^{CESM}$ are statistically more extreme than the seasons $\mathbb{X}^{ERAI}$.


Considering now the grid point 116°W/39°N in Nevada, USA, we find a substantially different ERA-Interim extreme summer substructure compared to eastern India (Figs. 3b,c), with largest extreme summer $RDA_{d,k}^{ERAI}$ values in the coldest third of the summer days and $XF_{cold}^{ERAI}$=44%, $XF_{middle}^{ERAI}$=29% and $XF_{hot}^{ERAI}$=27%. Also for this grid point, the substructure of CESM extreme summers is similar to that of ERA-Interim extreme summers, with $XF_{cold}^{CESM}$=45%, $XF_{middle}^{CESM}$=31% and $XF_{hot}^{CESM}$=24% (Fig.

4c). Thus, at this grid point, all thirds of the T2m distribution contribute to extreme summers, but the contribution from the coldest third is over proportionally large (i.e., considerably larger than 33%). Hence, the re-analysis and the climate model data both suggest that the suppression of cool summer days (leading to coldest days of the summer that are milder than usually) is a key ingredient for extreme summers at 116°W/39°N.

Yet a further extreme summer substructure is apparent at the grid point closest to Munich, Germany (12°E/48°N, Figs. 3d, 4d). At this grid point, the ERA-Interim extreme summers of 2017 and 2018 were characterized by $RDA_{d,k}^{ERAI}$-values of 1.5–2.5 K for all ranks, i.e., these summers resulted from an almost uniform shift of approximately 2 K in the entire T2m distribution.



Finally, the grid point 48°E/56°N in western Russia (Fig. 3e) illustrates that clearly distinct extreme summer substructures can

occur at the same grid point. While the extreme summer 2013 exhibited largest $RDA_{d,2013}^{ERAI}$ values for relatively cool summer

days ($SF_{cold,2013}^{ERAI}$=62%, $SF_{middle,2013}^{ERAI}$=28% and $SF_{hot,2013}^{ERAI}$=10%), the extreme summer 2010 was characterized by $RDA_{d,2010}^{ERAI}$

values in excess of 4 K for ranks ~40–92 ($SF_{cold,2010}^{ERAI}$=19%, $SF_{middle,2010}^{ERAI}$=35% and $SF_{hot,2010}^{ERAI}$=46%). The truly exceptional

nature of the 2010 summer in Russia at 48°E/56°N (e.g., Barriopedro et al. 2011, Fig. 3e) and of the 2003 summer in Central

Europe at 12°E/48°N (e.g., Stott et al. 2004, Fig. 3d) becomes particularly evident when comparing their $RDA_{d,k}^{ERAI}$ values with

those of the CESM extreme summers at the same grid points (Figs. 4d,e). For some ranks, none of the 700 CESM JJA seasons

reach $RDA_{d,k}^{CESM}$ values of comparable magnitude to those observed during the 2003 and 2010 summers at these two grid points.

Some implications of this finding will be discussed in Sect. 4.

In summary, the extreme summer substructure at these four grid points is qualitatively remarkably similar for the 5 hottest

ERA-Interim summers and the 35 hottest CESM summers. On the one hand, this similarity implies that the rank day anomaly

patterns presented in Figs. 3b-e are not artefacts of the rather short ERA-Interim period, but rather must result from physical

processes that shape the local extreme summer substructure. On the other hand, these similarities suggest that the CESM is

able to correctly capture the processes that generate the distinct extreme summer substructures at these example grid points.

We next compare the ERA-Interim and CESM extreme summer substructures at all grid points in the Northern Hemisphere

by considering the spatial patterns of $XF_{cold}^{ERAI}$, $XF_{hot}^{ERAI}$, $XF_{cold}^{CESM}$ and $XF_{hot}^{CESM}$.

### 3.3 Spatial variability of ERA-Interim and CESM extreme summer substructure

If extreme summers resulted from a uniform shift in the entire T2m distribution, all three thirds of the T2m distribution would

contribute equally (i.e., 33%) to $XA^{ERAI}$. However, the $XF_{hot}^{ERAI}$ field (Fig. 5a) reveals a complex pattern of coherent regions

with increased (> 33%) or decreased (< 33%) contributions from the hottest third of extreme summer days to $XA^{ERAI}$. Land

areas where particularly large $XF_{hot}^{ERAI}$ values are found include the central US, the UK, parts of northeastern Europe, India and

southeast Asia as well as the southern Sahel region (Fig. 5a). In some of these areas, $SF_{hot,k}^{ERAI}$ exceeded $SF_{middle,k}^{ERAI}$ and $SF_{cold,k}^{ERAI}$

during at least 4 out of 5 ERA-Interim extreme summers (stippling in Fig. 5a). In these regions, at least 4 out of 5 extreme

summers thus exhibited a similar substructure. Furthermore, also in parts of the northern North Pacific and northern North

Atlantic, $XF_{hot}^{ERAI}$ is substantially increased and reaches up to 60%. In many regions, however, $XF_{hot}^{ERAI}$ is less than 33%,

indicating that in these regions, extreme summers do not arise primarily from the hottest 30 days of the summer being hotter

than climatologically.

In fact, in many regions it is the contribution to $XA^{ERAI}$ from the coldest third of the summer ($XF_{cold}^{ERAI}$) that is substantially

increased (Fig. 5c), for example the southwestern US, the northern Sahel region, Pakistan and parts of Greenland. Moreover,



increased $XF_{cold}^{ERAI}$ values are also found in the southern North Pacific and the southern North Atlantic as well as over the Arctic Ocean (Fig. 5c). Overall, Fig. 5c clearly demonstrates that the coldest third of all summer days contributes a substantial fraction to $XA^{ERAI}$ in most regions. In fact, in 46% of the Northern Hemisphere land area, $XF_{cold}^{ERAI}$ exceeds $XF_{hot}^{ERAI}$, i.e., the coldest third of extreme summers contributes more to $XA^{ERAI}$ than the hottest third. Consequently, in these regions the

mechanisms that suppress unusually cool summer days must be considered when assessing the physical causes of extremely hot summers.

Comparing these results derived from ERAI with results based on CESM, i.e., $XF_{hot}^{ERAI}$ and $XF_{hot}^{CESM}$ (Figs. 5a,b) as well as $XF_{cold}^{ERAI}$ and $XF_{cold}^{CESM}$ (Figs. 5c,d), unravels strikingly similar patterns in many regions. For example, both data sets agree (even

quantitatively) that extreme summers in India and Southeast Asia come about primarily by the hottest summer days being hotter than climatologically, while the coldest third of extreme summer days only contributes a marginal fraction to the respective $XA$. Also in the western and central US, $XF_{cold}$ and $XF_{hot}$ agree very well between the two data sets, with the cool summer days contributing an over proportionally large fraction to $XA$ in the western US, and the hot summer days in the central US. Further areas of remarkable agreement between $XF_{cold}^{ERAI}$ and $XF_{cold}^{CESM}$ (Figs. 5c,d) are the high Arctic and the northern

Sahel region. Moreover, in 50% of the Northern Hemisphere land area $XF_{cold}^{CESM}$ exceeds $XF_{hot}^{CESM}$, which compares well with the 46% of the land area in which $XF_{cold}^{ERAI}$ exceeds $XF_{hot}^{ERAI}$. Figure 5 thus clearly reveals that the CESM reproduces many features of the observed extreme summer substructure and its variability in space to a remarkable degree.

However, there are also some areas of notable differences between $XF_{hot}^{ERAI}$ and $XF_{hot}^{CESM}$ as well as $XF_{cold}^{ERAI}$ and $XF_{cold}^{CESM}$. For

example over Greenland, Saudi Arabia and the northern North Atlantic, there are substantial differences between $XF_{cold}^{ERAI}$ and $XF_{cold}^{CESM}$ (Figs. 5c,d). Moreover, over the northern North Pacific as well as the high Arctic, the $XF_{hot}^{CESM}$ and $XF_{hot}^{ERAI}$ patterns agree only qualitatively, but not quantitatively (Figs. 5a,b). It is important to note, though, that some differences in the $XF_{cold}$ and $XF_{hot}$ fields for the two data sets are expected due to the different sample sizes, even if the model was perfect. In the remainder of this paper we aim to explain statistical and physical reasons behind selected aspects of the spatial variability in

$XF_{cold}$ and $XF_{hot}$.

### 3.4 A statistical explanation for the observed extreme summer substructures

Figures 3b,c and 4b,c clearly illustrate that, at the selected grid points in India (81°E/21°N) and in the US (116°W/39°N) some rank days are climatologically much more variable than others. Importantly, this is the case not just for extreme summers but

it is rather a climatological characteristic of the local temperature variability. For example, at 81°E/21°N the hottest 30 days of the summer are much more variable than the colder days. The 5th to 95th percentile range of the $RDA_{80,k}^{CESM}$-values is roughly 5 times larger than that of the $RDA_{10,k}^{CESM}$-values (Fig. 4b). At 116°W/39°N the largest rank day variability is found for lower





ranks and the 5$^{th}$ to 95$^{th}$ percentile range of the $RDA_{80,k}^{CESM}$ values is roughly 2 times smaller than the same percentile range of the $RDA_{10,k}^{CESM}$-values (Fig. 4c). Similar ratios are found when comparing the spread of $RDA_{80,k}^{ERAI}$ and $RDA_{10,k}^{ERAI}$ for these two

grid points (Figs. 3b,c). Moreover, at both grid points extreme summers occur when the most variable rank days are particularly hot (Figs. 3b,c and 4b,c). Hence, from a statistical point of view, the extreme summer substructure at these two particular grid points appears to be largely determined by the local "rank day variability pattern". That is, the contributions to $XA$ from the distinct rank days during extreme summers depend on how variable the respective values $T_{d,k}$ are climatologically.


We next assess whether the local rank day variability pattern also explains the extreme summer substructure at other Northern Hemisphere grid points. To do so, we consider the variance ($V$) of the $RDA_{d,k}$ values of all ranks and all JJA seasons at a particular grid point:

$$V = \frac{1}{K \cdot D} \sum_{k=1}^{K} \sum_{d=1}^{D} (RDA_{d,k})^2. \tag{7}$$

Here we have used the fact that the mean of the $RDA_{d,k}$ values is by construction equal to zero and thus their variance reduces

to the average of the squared $RDA_{d,k}$-values of all $d$ and all $k$. The contributions from the coldest, middle and hottest third to $V$ are then e.g.,

$$VF_{cold} = \left( \frac{1}{K \cdot D} \sum_{k=1}^{K} \sum_{d=1}^{\left[\frac{D}{3}\right]} (RDA_{d,k})^2 \right) \Big/ V, \tag{8}$$

and analogously for the middle and hottest third of the summer days.

The fields of $V^{ERAI}$ and $V^{CESM}$ (Figs. 6a, 7a) closely resemble the $XA^{ERAI}$ and $XA^{CESM}$-fields (Figs. 3a, 4a), as large rank day

anomalies are a prerequisite for large seasonal T2m anomalies. Furthermore, comparing $XF_{hot}^{ERAI}$ and $VF_{hot}^{ERAI}$ (Figs. 5a and 6b) clearly reveals that wherever the contribution from the hottest third of the summer days to $XA^{ERAI}$ is increased ($XF_{hot}^{ERAI} >$ 33%), the rank day variability in the hottest third (quantified by $VF_{hot}^{ERAI}$) contributes over proportionally to $V^{ERAI}$. Figures 5c and 6c illustrate that the same relationship also holds for $XF_{cold}^{ERAI}$ and $VF_{cold}^{ERAI}$: regions where milder than normal cool summer days contribute over proportionally to $XA^{ERAI}$ (i.e., $XF_{cold}^{ERAI} >$ 33%) exhibit increased $VF_{cold}^{ERAI}$ values. Figures 5b,d and 7b,c

confirm this finding also for the CESM data. We thus conclude that in both data sets, the extreme summer substructure is largely determined by the local rank day variability pattern.

Furthermore, comparing the patterns of $VF_{hot}^{ERAI}$ and $VF_{hot}^{CESM}$ (Figs. 6b, 7b) reveals agreement in the same regions where also the patterns of $XF_{hot}^{ERAI}$ and $XF_{hot}^{CESM}$ (Figs. 5a,b) agree, and, conversely, disagreement between $VF_{hot}^{ERAI}$ and $VF_{hot}^{CESM}$ also results





in disagreement between $XF_{hot}^{ERAI}$ and $XF_{hot}^{CESM}$. For example, the $VF_{hot}^{ERAI}$ and $VF_{hot}^{CESM}$ fields (and the $XF_{hot}^{ERAI}$ and $XF_{hot}^{CESM}$ fields) are almost identical in India and Southeast Asia, the northern Sahel, the western US or Eastern Europe (cf. Figs. 6b and 7b, and Figs. 5a,b). Over Saudi Arabia or the northern North Atlantic, however, the patterns of $VF_{hot}^{ERAI}$ and $VF_{hot}^{CESM}$ (and of $XF_{hot}^{ERAI}$ and $XF_{hot}^{CESM}$) do not agree particularly well. In summary, while the CESM correctly reproduces the local rank day variability pattern in most regions, differences in the local rank day variability patterns between the two data sets also lead to

differences in the extreme summer substructures.

It is interesting to compare the $FV_{cold}$ and $FV_{hot}$ patterns presented in Figs. 6 and 7 with the skewness of the local daily temperature distributions, which has been studied extensively in the past (Donat and Alexander, 2012; Garfinkel and Harnik, 2017; Linz et al., 2018; Loikith et al., 2018; Loikith and Neelin, 2015; Ruff and Neelin, 2012). The upper tail of, e.g., a

positively skewed JJA T2m distribution is longer than the lower tail, which is the case if the hottest summer days are more variable than the coldest summer days. Hence, explanations of distinct skewness in daily T2m distributions also help to understand differences in the rank day variability patterns and, subsequently, extreme summer substructures. Garfinkel and Harnik (2017) showed that the winter low-level temperature distributions are positively skewed on the cold side of the Northern Hemisphere storm tracks, primarily because there the magnitude of warm air advection exceeds that of cold air advection.

And, vice versa, the winter low-level temperature distributions are negatively skewed on the warm side of the Northern Hemisphere storm tracks, where the magnitude of cold air advection exceeds that of warm air advection. Consistent with their results, Figs. 6 and 7 depict more variable hot summer days to the north and more variable cold summer days to the south of the Northern Hemisphere storm tracks, where the horizontal gradients of T2m are particularly large (see in particular yellow contours in Figs. 6b,c).


While this argument explains differences in the rank day variability and the extreme summer substructures in regions of strong surface temperature gradients, Figs. 5-7 also reveal numerous rather small-scale features, that do not necessarily occur in regions of strong surface temperature gradients. We therefore next analyze the extreme summer substructure and its causes in three example regions in more detail. Due to the similarity between the ERA-Interim and CESM extreme summer

substructures, we restrict this analysis to ERA-Interim data (except where mentioned otherwise).

### 3.5 (Examples of) physical causes of extreme summer substructures

A particularly striking feature of Fig. 5 is the large contribution from the hottest third of the summer days to $XA^{ERAI}$ in India, illustrated exemplarily for the grid point at 81°E/21°N in Fig. 3b. The general temperature evolution in JJA (i.e., considering

all JJA seasons) at this grid point follows a particular sub-seasonal pattern (Fig. 8a). In early June, ERA-Interim T2m values are highly variable and range from 27°C to almost 40°C, with a mean of 35°C on 1 June. Throughout June and the first half of July the climatological T2m drops to approximately 26°C and remains at this level until the end of August. Moreover, during



that period, the variability in T2m is much smaller than in early June. The extreme summers exhibit comparatively high temperatures primarily in June, while in July and August their T2m evolution does not differ substantially from other JJA

seasons (Fig. 8a). The drop of T2m in June is associated with the onset of the Indian summer monsoon [Fig. 8b; e.g., Slingo, (1999)]. During most JJA seasons, precipitation starts to fall already during the first half of June. However, the extreme summers each featured very little precipitation for at least the first 20 days of June, which suggests that extreme summers at this grid point occur when there is an unusually late onset of the Indian summer monsoon at this particular location. Moreover, the rank day variability pattern at 81°E/21°N is easily understood from Fig. 8: The hottest days of the season mostly occur in

June and are associated with dry conditions. The onset date of the monsoon determines how many dry (and thus very hot) days occur in a JJA season, i.e., an early onset of the Indian monsoon suppresses a large number of very hot days and a late onset increases this number, which leads to the large temperature variability seen in the warmest 30 days of the JJA season.

A further noteworthy feature in Fig. 5 is the sharp boundary in the extreme summer substructure around 75°N–80°N, for

example in the North Atlantic sector. North of this boundary, the coldest third of all extreme summer days contribute up to 60% to the extreme summer anomaly (Figs. 5c,d). South of it, the contribution from the coldest third of extreme summer days is much smaller. (Quantitatively, there is some disagreement between the CESM and ERAI extreme summer substructures, but both data sets agree about the general pattern.) This sharp boundary in the extreme summer substructure is co-located with the climatological sea ice edge in JJA (Fig. 9a). Examining the JJA T2m distributions at three grid points across this boundary

(42°W/83°N, 42°W/81°N and 42°W/79°N) reveals that for T2m below –1°C, their probability density functions (pdfs) of the daily T2m values are almost identical, which is not surprising due to their close spatial proximity. However, large differences in the three pdfs are found for T2m at about 0°C and above. At 83°N, i.e., north of the climatological sea ice edge (Fig. 9a), the pdf exhibits a very short upper tail with very little probability density exceeding +2°C (i.e., the pdf is strongly negatively skewed), while at 79°N (i.e., south of the climatological sea ice edge) the upper tail is much more variable. The geographical

co-location of this extreme summer substructure boundary and of the climatological sea ice edge is striking and suggests that the contrasting substructures arise because the sea ice buffers "warm" temperatures at 0°C, that is, air with T2m > 0°C is cooled down to close to 0°C by the induced sea ice melting. The same effect has also been shown to shorten the upper tail of the surface temperature pdf over snow covered areas (Loikith et al., 2018).

As a third example, we return to the grid point in Nevada, US (at 116°W/39°N), where the rank day variability is largest for the cold summer days and extreme summers occur when the coldest 30 days exhibit mostly large positive rank day anomalies (Figs. 3c and 4c). Thus, at this grid point, milder than normal coldest days of the summer (or, equivalently, suppressed cool summer days) are a key ingredient for extreme summers. We therefore briefly explore why, at this grid point, the coldest summer days during extreme summers are warmer than normally.






We first investigate what makes the climatologically coldest summer days at 116°W/39°N particularly cold and then contrast them with the coldest summer days during extreme summers at 116°W/39°N. A composite analysis of the upper-level flow during the 100 climatologically coldest ERA-Interim days of all 1979–2018 summers unravels a characteristic upper-level flow pattern: a highly amplified Rossby wave pattern over the eastern North Pacific and North America, with a breaking

synoptic-scale trough covering 116°W/39°N (Fig. 10a). The breaking Rossby wave causing the trough is part of a synoptic-scale and transient wave packet (Fig. 10b) which has just the right phasing such that the trough axis crosses 116°W/39°N when the amplitude of the trough is largest (Fig. 10b). This type of relatively small-scale troughs, shown here with contours of potential vorticity on an isentrope in the upper troposphere (Fig. 10a), is relatively slow moving (Fig. 10b), such that the induced northwesterly low-level flow along its western flank can lead to strong and persistent cold-air advection to the western

US. Additionally, the  low-level flow induced by the trough impinges on the topography at the US west coast. Consequently, low-level air masses that are advected into the western US are most likely forced to ascend, which leads to adiabatic cooling of these already cool airmasses and finally results in the climatologically coldest summer days at 116°W/39°N.

The composites for the 100 coldest days during extreme summers, in contrast, do not reveal such a wave pattern (Figs. 10a

and 10c). This indicates that the flow pattern characteristic of the climatologically coldest days at this grid point, i.e., the Rossby wave breaking and trough formation with the phasing discussed above, simply did not occur very often during extreme summers. Furthermore, a synoptic analysis of these 100 coldest extreme summer days (not shown) reveals that the associated upper-level flow configurations are rather variable, some featuring troughs while others even exhibited low-amplitude ridges, resulting in the rather zonal composite upper-level flow apparent in Figs. 10a and 10c.


Why in extreme summers at 116°W/39°N such highly amplified troughs with the right phasing did not occur is currently unclear, and at the same time challenging to assess. Possibly, the exact longitude where the synoptic-scale waves have been triggered (Röthlisberger et al., 2018) as well as the strength and longitudinal extent of the North Pacific jet, which modulates the waves' downstream propagation and breaking behavior (e.g., Drouard et al. 2015), might have played a role. However,

both the jet strength and the characteristics of the transient waves propagating along the jet are strongly modulated by lower-frequency processes such as the Madden-Julian Oscillation (Moore et al., 2010) and the El Niño Southern Oscillation (Drouard et al., 2015; Shapiro et al., 2001). This example thus illustrates that a seamless approach, combining processes on different time scales, is most likely required to fully reveal the physical causes of extreme summers.

## 4    Summary and concluding remarks

In this study, extreme summers are defined in the upper tail of the JJA seasonal mean T2m distribution at each grid point in the Northern Hemisphere and then analyzed with regard to their substructure. Hereby, the extreme summer T2m anomaly is decomposed into its contribution from each rank day. First, all days are ranked within their respective season (i.e., from rank



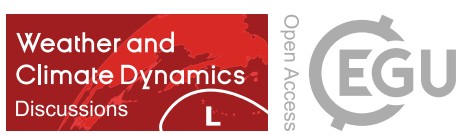

1 to 92 for JJA) and then compared to the climatological T2m of all days with the same rank. The resulting rank day anomalies exactly quantify how much each (rank) day contributes to the T2m anomaly of the respective season. This decomposition is

applied to T2m data from ERA-Interim as well as data from 700 simulated years with CESM for present day climate conditions. Thereby, the contributions from the coldest, middle and hottest third of extreme summers to the extreme summer T2m anomalies are quantified at each Northern Hemisphere grid point ($XF_{cold}$, $XF_{middle}$ and $XF_{hot}$).

This analysis reveals clearly distinct extreme summer substructures, occurring in coherent geographical regions. Despite the

relatively small scale of the structures in the $XF_{cold}^{ERAI}$ and $XF_{hot}^{ERAI}$ fields as well as different numbers of extreme summers in the two data sets, CESM is able to reproduce these fields to a remarkable degree. This result firstly underlines that the ERA-Interim extreme summer substructures and their spatial variability result from physical processes rather than a too short data record and, secondly, testifies to the model's ability to reproduce the physical processes responsible for the occurrence of extreme summers in most regions in the Northern Hemisphere. Areas where CESM and ERA-Interim extreme summer

substructures differ include Greenland, the northern North Atlantic as well as the Arabian Peninsula.

Furthermore, we show that the extreme summer substructure is largely determined by which of the 92 JJA rank days are most variable (i.e., the rank day variability pattern). Simply speaking, in regions where the coldest days of the summer are most variable, extreme summers occur when the coldest days of the summer are unusually hot, and analogously for the case where

hottest days vary the most. Moreover, the few areas where the ERA-Interim and CESM extreme summer substructures differ, also have distinct rank day variability patterns in ERA-Interim and CESM. Thus, the climate model's ability to reproduce the ERA-Interim extreme summer substructures in most places results largely from the model's ability to produce local rank day variability patterns that agree with ERA-Interim.

The rank day variability pattern is qualitatively related to the skewness of the T2m distribution: regions with positively (negatively) skewed T2m distributions exhibit larger variability in the upper (lower) ranks [e.g., compare Figs. 6 and 7 of this study with Fig. 4 in Loikith et al. (2018)]. Thus, for interpreting our results, we partly rely on previous work that investigated the physical causes of skewed surface temperature distributions. Consistent with the findings of Garfinkel and Harnik (2017), we find distinct extreme summer substructures relative to the location of large surface temperature gradients, in particular in

the Northern Hemisphere storm track regions. Extreme summers occurring north of the Northern Hemisphere storm tracks have large contributions from the hottest third of summer days, and south of the storm tracks the contributions from the coldest days are largest. This is primarily because on the cold side of a temperature gradient, warm air advection can reach much larger magnitudes than cold air advection, and vice versa on the warm side (e.g., Garfinkel and Harnik, 2017).

However, three case studies illustrate that the extreme summer substructure often cannot be explained by temperature advection alone. In eastern India, more than 70% of the extreme summer T2m anomaly results from the hottest 30 days of JJA





being hotter than climatologically. At the considered grid point, T2m exhibits a distinct sub-seasonal pattern, as it typically drops by almost 10 K with the onset of the Indian summer monsoon. Thus, the hottest days of the season (occurring in June) are highly variable, and extreme summers occur in seasons with particularly late monsoon onsets.


In the high Arctic the highest surface temperatures are buffered around 0°C, as excess heat would result in sea ice melting and subsequent latent cooling. Hence, the cold part of the T2m distribution accounts for most of the rank day anomaly variance and, consequently, extreme summers occur when the coldest summer days are warmer than normally. This buffering effect of the Arctic sea ice leads to a strong boundary in the extreme summer substructure around 75°N-80°N, i.e., near the

climatological JJA sea ice edge.

At a grid point in the western United States, all parts of the T2m distribution contribute significantly to extreme summers, however, an over proportionally large fraction comes from the coldest third of the extreme summer days (i.e., the coldest extreme summer days are warmer than their rank day mean). Composites of the upper-level flow during the 100

climatologically coldest summer days reveal that an amplified upper-level flow pattern with a particular phasing of a prominent trough and its associated cold air advection is characteristic of the climatologically coldest summer days at this grid point. This particular flow pattern did not occur frequently during the extreme summers, leading to milder than normal cool summer days. Overall, the case studies illustrate that for understanding the physical causes of extreme summers, a seamless approach is necessary, which combines weather system dynamics, local thermodynamics and surface-atmosphere interactions as well as

lower frequency variability in the atmosphere and the ocean. The extreme summer substructure is hereby a helpful tool for identifying coherent regions with similar physical causes of extreme summers.

A key result of this study is that in most places, the cool summer days contribute substantially to extreme summer T2m anomalies. In fact, Fig. 5 reveals that for ERA-Interim (CESM) in 46% (50%) of the Northern Hemisphere land area, the

coldest third of the summer contributes more to the extreme summer anomaly ($XA$) than the hottest third. Thus, whenever large positive seasonal temperature anomalies are of interest (i.e. extreme summers as opposed to individual heat waves), the suppression of cold summer days is fundamentally important. Yet, the responsible processes are so far virtually unexplored and thus possibly yield an untapped potential for improving our understanding of extreme summers. However, as illustrated by the example of extreme summers in the western US, the processes that suppress the occurrence of cold summer days

sometimes seem rather intangible, as they often manifest themselves in the non-occurrence of a particular flow pattern rather than the occurrence of the flow pattern.

This study has illustrated that extreme summers across the Northern Hemisphere have distinct substructures, which result directly from the physical causes of the extreme summers. However, the concept of the extreme season substructure has

applications beyond what has been presented in this study and thus calls for subsequent studies. Firstly, the presented analyses





could be extended to the Southern Hemisphere and other seasons and variables. (The application of the technique is most promising for variables that are potentially unbound and variable on both ends, i.e., not for a positive definite variable like precipitation.) Secondly, extreme summers with distinct substructures conceivably have different societal effects and thus future research should assess whether or not and where the extreme summer substructure is affected by climate change. The

results of this study suggest that the CESM is a suitable tool for this task, as it is largely able to reproduce the observed (ERA-Interim) extreme summer substructure in the current climate. However, some of the extreme summers observed within the last 40 years appear to be outside of the spectrum of 700 years of CESM. Hence, while CESM is able to reproduce the local extreme summer substructures, it may not be able to reproduce the most extreme summers that are physically possible in some regions. Clearly, this finding requires detailed and critical further investigation. Finally, changes in the extreme summer substructure

with climate change must be related to changes in the physical causes of extreme summers, as a uniform warming would not affect the local rank day variability pattern. Therefore, contrasting extreme summer substructures in present and future climate simulations might also help to identify regions where the physical causes of extreme summers are altered by climate change.

*Data availability.* ERA-Interim data can be downloaded from the ECMWF webpage

(https://apps.ecmwf.int/datasets/data/interim-full-daily/levtype=sfc/). The CESM T2m data used here is available upon request from the authors.

*Author contributions.* MR and HW conceived the study, MS provided technical support, UB performed the CESM simulations, MR analyzed the data and wrote the major part of the manuscript. HW, EF, MS, and UB also contributed to

writing the manuscript and commented on earlier versions of this manuscript.

*Competing interests.* The authors declare no conflict of interest.

*Acknowledgements.* MR, EF and HW acknowledge funding of the INTEXseas project from the European Research Council

(ERC) under the European Union's Horizon 2020 research and innovation programme (grant agreement No 787652). Moreover, Maxi Boettcher (ETH Zürich) and Lukas Papritz (ETH Zürich) are acknowledged for helpful discussions during different stages of this work, and Gary Strand (NCAR) and Clara Deser (NCAR) for providing the CESM restart files.



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

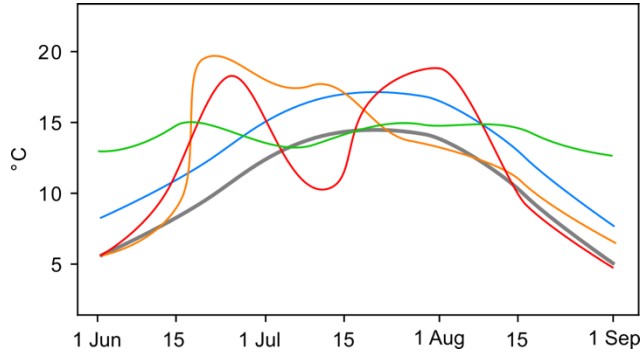

**Figure 1.** Schematic surface temperature evolution during extreme summers with different substructures: an extreme summer arising from just one heat wave (orange), from two heat waves, with a rather cool period separating the two heat waves (red), from a suppression of cool summer days (green) and from a shift in the entire T2m distribution (blue). The schematic climatological surface temperature

evolution is depicted in gray.





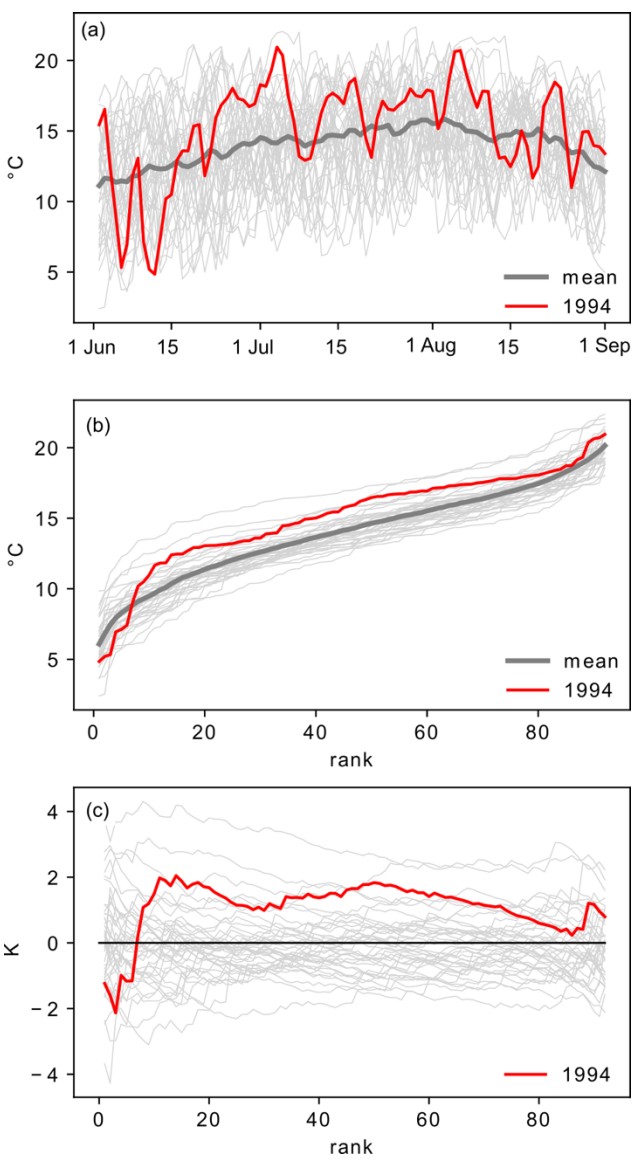

**Figure 2.** Steps in computing $RDA_{d,k}^{ERAI}$-values at the grid point closest to Zürich, Switzerland (9°E/47°N). Values for the 1994 summer are highlighted in red. Panel (a) shows ERA-Interim T2m at 9°E/47°N for all 40 ERA-Interim summers. The sorted T2m values ($T_{d,k}^{ERAI}$) are shown in panel (b) and the $RDA_{d,k}^{ERAI}$-values in panel (c).






**Figure 3.** Extreme summer T2m anomaly and extreme summer substructure for selected grid points in ERA-Interim. Panel (a) depicts $XA^{ERAI}$, panels (b–e) show $RDA_{d,k}^{ERAI}$ for the five ERA-Interim extreme summers in colours and for the remaining summers in light grey. Crosses in panel (a) indicate the grid points for which the $RDA_{d,k}^{ERAI}$-values are shown in panels (b–e).





**Figure 4.** Extreme summer T2m anomaly and extreme summer substructure for selected grid points in CESM. Panel (a) displays $XA^{CESM}$ and panels (b–e) show in red the maximum and minimum (dotted), 90th and 10th percentile (dashed) and the median (solid red) $RDA^{CESM}_{d,k}$ of the 35 CESM extreme summers. The 5th to 95th percentile range of the $RDA^{CESM}_{d,k}$ of all JJA seasons are depicted in grey. Crosses in panel (a) indicate the grid points for which the rank day anomalies are shown in panels (b–e).

**Figure 5.** Spatial variability in the extreme summer substructure in ERA-Interim and CESM. Panels (a) and (b) depict $XF_{hot}^{ERAI}$ and $XF_{hot}^{CESM}$, respectively, while $XF_{cold}^{ERAI}$ and $XF_{cold}^{CESM}$ are shown in panels (c) and (d). Stippled areas in all panels indicate grid points at which the same third of the distribution contributes the largest fraction of all thirds to at least 80% of the extreme summers (i.e., similar substructure in at least 80% of the extreme summers). Black crosses as in Fig. 3a.



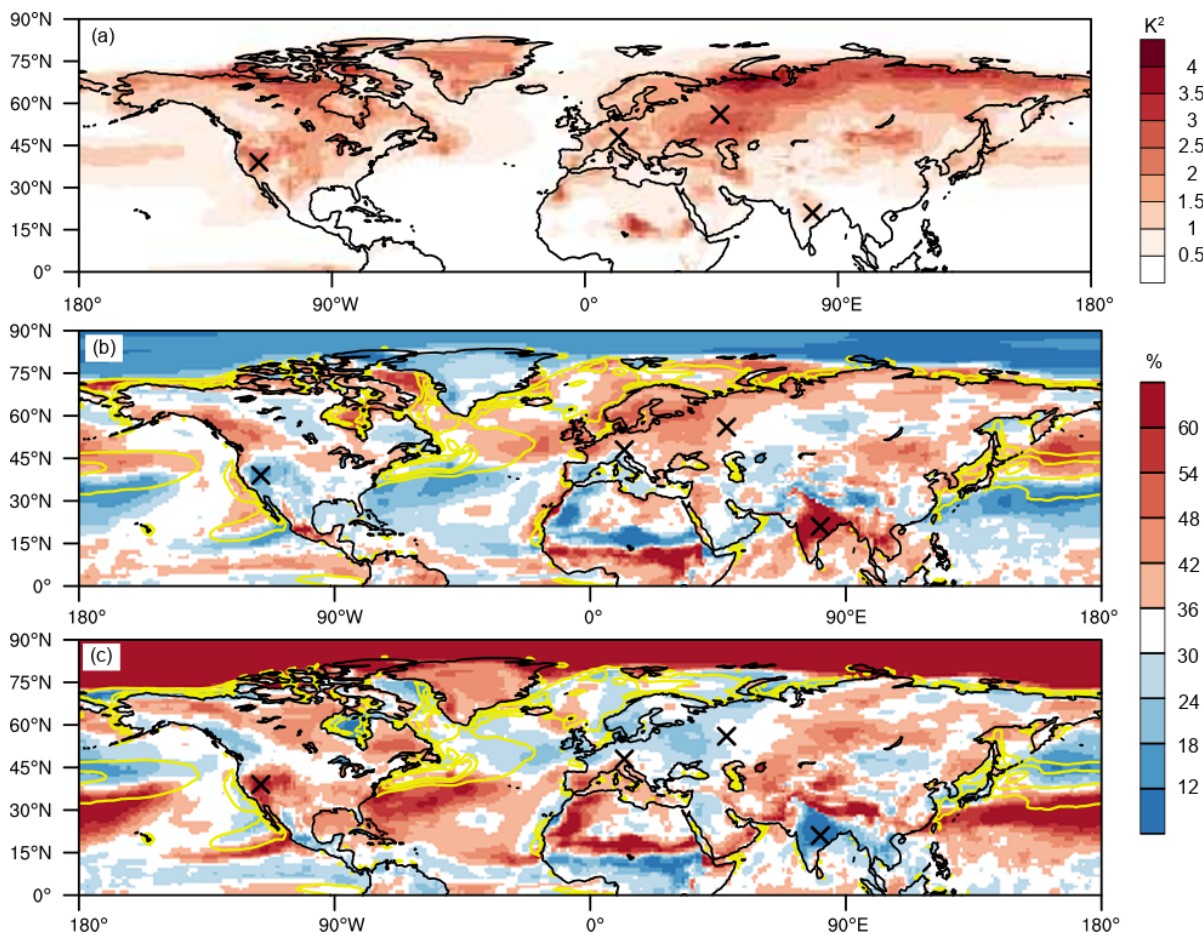

**Figure 6.** The variance of $RDA_{d,k}^{ERAI}$ and its contributions from the coldest and hottest third of summer days. Panel (a) depicts $V^{ERAI}$ and panels (b) and (c) show $VF_{hot}^{ERAI}$ and $VF_{cold}^{ERAI}$, respectively. Yellow contours in (b) and (c) depict $C^{ERAI}$ gradient magnitudes of 5, 10 and 15 K $10^6$ m$^{-1}$. The $C^{ERAI}$ gradient magnitudes have been computed as first order central differences and are only plotted over oceans. Black crosses as in Fig. 3a.





**Figure 7.** The variance of $RDA_{d,k}^{CESM}$ and its contributions from the coldest and hottest third of summer days. Panel (a) depicts $V^{CESM}$ and panels (b) and (c) show $VF_{hot}^{CESM}$ and $VF_{cold}^{CESM}$, respectively. Black crosses as in Fig. 3a.





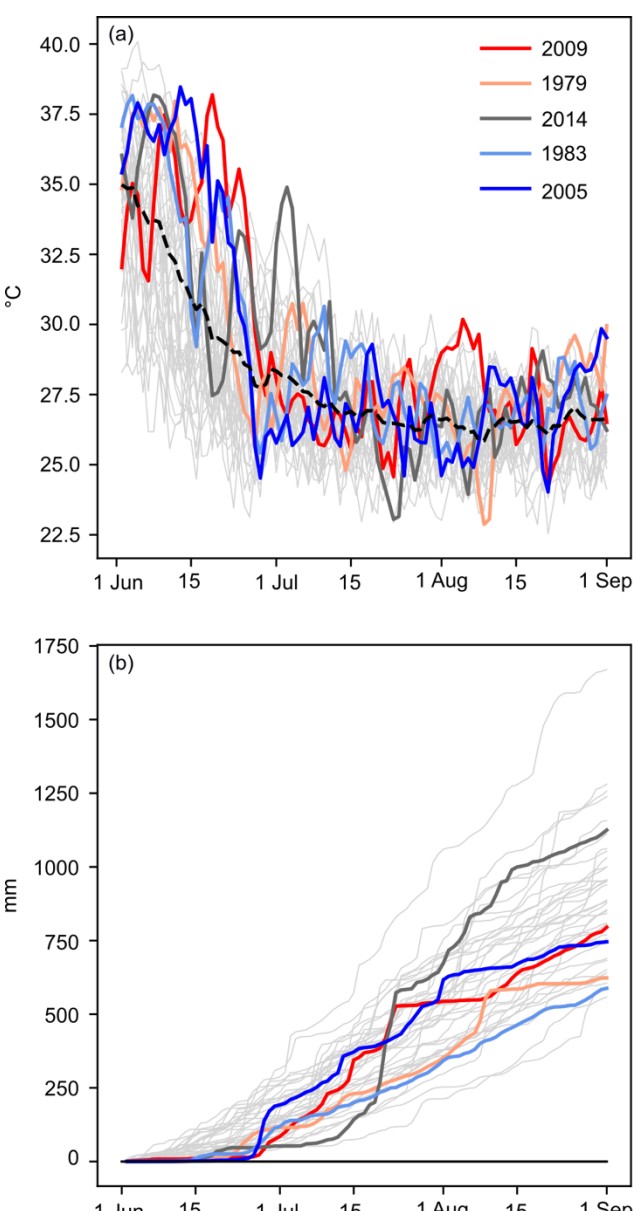

**Figure 8.** The JJA temperature and precipitation evolution at 81°E/21°N. Panels (a) and (b) depict ERA-Interim T2m and accumulated
precipitation at 81°E/21°N for all JJA seasons, respectively. The extreme summers are highlighted in colors. The dashed black line in (a)
depicts the climatological calendar day mean T2m at 81°E/21°N.





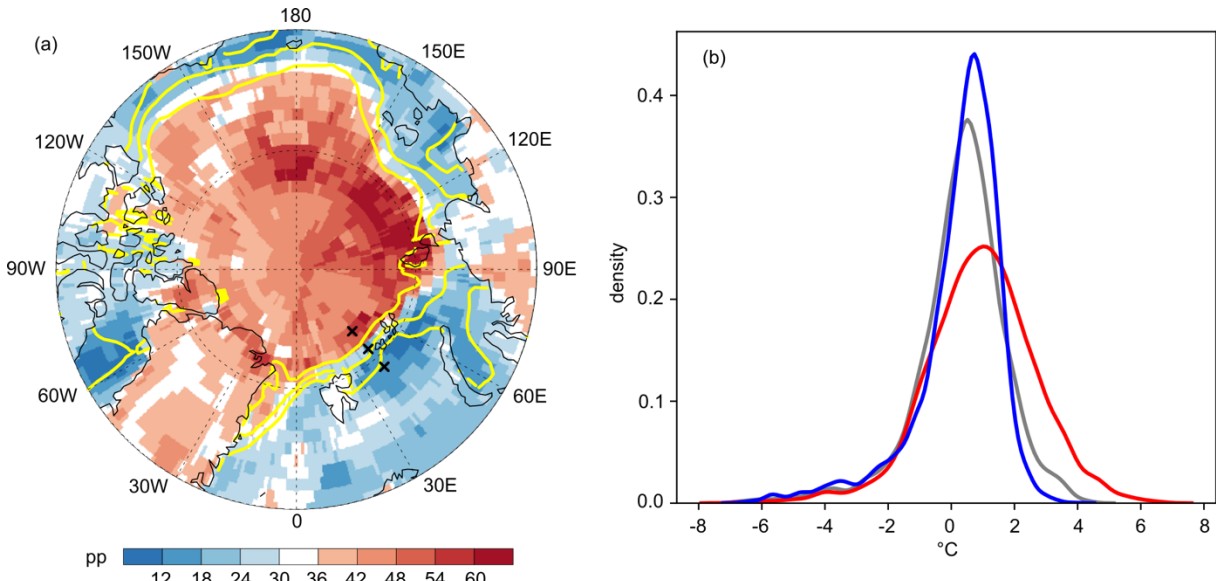

**Figure 9.** Arctic sea ice and local summer temperature variability. Panel (a): $XF_{cold}^{ERAI}$ (shading, only 70°N–90°N is shown) and mean 1979–2018 JJA ERA-Interim sea ice concentration (yellow contours indicate sea ice concentrations of 0.3, 0.5 and 0.7). Panel (b): empirical probability density function of ERA-Interim T2m at 79°N/42°E (red), 81°N/42°E (gray) and 83°N/42°E (blue). Crosses in (a) locate these three grid points.





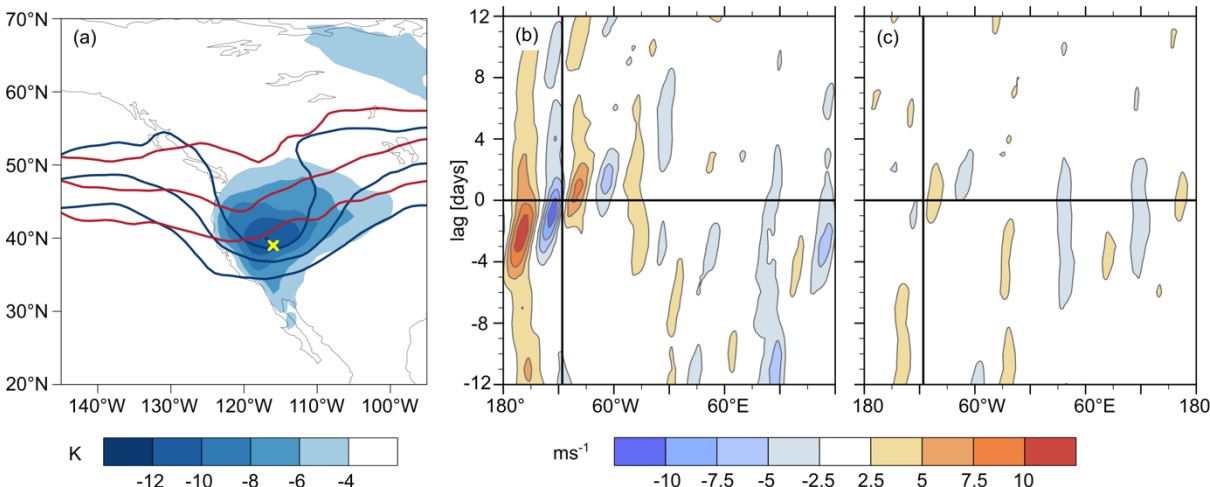

**Figure 10.** (a) T2m difference between the 100 climatologically coldest JJA days and the 100 coldest extreme summer days (shading). Contours depict the composite PV field at 335 K (contours of 2, 3.5 and 5 PVU) for the 100 climatologically coldest JJA days (blue) and for the 100 coldest extreme summer days (red). The yellow cross indicates 116°W/39°N. Panels (b) and (c) depict composite Hovmöller diagrams of the anomalous 250 hPa meridional wind, averaged between 35°N and 65°N temporally centered on the 100 climatologically coldest JJA days (b) and on the 100 coldest extreme summer days (c). Meridional wind anomalies are calculated relative to the 1979–2018 mean JJA meridional wind. The vertical line in (b) and (c) indicates 116°W.





**Table 1.** Definitions and descriptions of important quantities used in this study.

| Symbol | Formal definition | Description |
|---|---|---|
| $T_{d,k}$ | | Daily mean T2m with rank $d$ in season $k$ (Fig. 2b) |
| $SM_k$ | $\dfrac{1}{D}\displaystyle\sum_{d=1}^{D} T_{d,k}$ | Seasonal mean T2m of season $k$ |
| $C$ | $\dfrac{1}{K \cdot D}\displaystyle\sum_{k=1}^{K}\sum_{d=1}^{D} T_{d,k}$ | Climatological JJA seasonal mean |
| $SA_k$ | $SM_k - C$ | Seasonal anomaly of season $k$ |
| $RDM_d$ | $\dfrac{1}{K}\displaystyle\sum_{k=1}^{K} T_{d,k}$ | Rank day mean of rank $d$ |
| $RDA_{d,k}$ | $T_{d,k} - RDM_d$ | Rank day anomaly of rank $d$ in season $k$ (Figs. 2c, 3b–e, 4b–e) |
| $XM$ | $\dfrac{1}{N}\displaystyle\sum_{k\in\mathbb{X}} SM_k$ | Mean of $N$ considered extreme summers |
| $XA$ | $XM - C$ | Mean anomaly of $N$ considered extreme summers (Figs. 3a, 4a) |
| $SF_{cold,k}$ | $\left(\dfrac{1}{D}\displaystyle\sum_{d=1}^{\left[\frac{D}{3}\right]} RDA_{d,k}\right)\Big/ SA_k$ | Fractional contribution from the coldest third of summer days of season $k$ to $SA_k$ |
| $XF_{cold}$ | $\left(\dfrac{1}{N}\displaystyle\sum_{k\in\mathbb{X}}\dfrac{1}{D}\sum_{d=1}^{\left[\frac{D}{3}\right]} RDA_{d,k}\right)\Big/ XA$ | Fractional contribution from coldest third of extreme summer days to $XA$ (Fig. 5) |
| $V$ | $\dfrac{1}{K \cdot D}\displaystyle\sum_{k=1}^{K}\sum_{d=1}^{D} (RDA_{d,k})^2$ | Variance of all $RDA_{d,k}$ values at a particular grid point. (Figs. 6a, 7a) |
| $VF_{cold}$ | $\left(\dfrac{1}{K \cdot D}\displaystyle\sum_{k=1}^{K}\sum_{d=1}^{\left[\frac{D}{3}\right]} (RDA_{d,k})^2\right)\Big/ V$ | Fractional contribution from the coldest third of all summer days to $V$ (Figs. 6b,c, 7b,c) |

655