# Peer review of "The substructure of extremely hot summers in the Northern Hemisphere"

_Weather and Climate Dynamics, 2019_

## Referee Comment (RC1) · Anonymous Referee #1 · 27 Dec 2019

**General comments**

This manuscript investigates Northern Hemisphere extreme hot summer seasons from a statistical point of view. The topic is relevant, because hot summers have societal impact and are going to become more frequent due to anthropogenic climate change. The paper focuses on the entire 3-month summer season rather than addressing individual heat waves (which have been studied before quite extensively). The method involves a novel statistical analysis based on ranking the 92 days of a summer season according to their anomaly with respect to the corresponding climatology.

The results indicate that hot summers in different areas on the Northern Hemisphere may have different substructure: in some regions a summer season tends to be hot because the hottest tercile is anomalously hot, while in other regions the summer sea-

son tends to be hot because the coldest tercile is anomalously hot. In addition, it is shown that the Community Earth System Model (CESM) is able to broadly represent such regional differences. The regional differences are made plausible by studying a few cases/locations. I think these are interesting results. In addition, the paper is very well written. I have a few specific comments below which may help to produce a final version.

My only general comment is the following. I found that the statistical method is well described and sounds very interesting, and while reading I was eagerly awaiting the discussion of possible physical causes. But then (reading that section) I was somewhat disappointed. For instance, the shift in the onset of the Indian monsoon obviously explains the behavior found in the statistical analysis; actually, the explanation is so obvious that in retrospect the statistical analysis almost appears as an artifact. Let me grossly exaggerate to make my point clear: if you have a very simple phenomenon and apply a rather complex or strange analysis to it, you are likely to find a complex or strange result, but the complexity or strangeness of the result in this case would be mostly a feature of the analysis an not a feature of nature. Having said this, I still believe that the analysis is worth doing, and you do it very well.

**Specific comments**

Line 68: Can you give here an example, too?! You could, for instance, mention Nevada (USA) and say that this will be discussed later.

Line 96: Do you really "illustrate physical causes"? I feel that you, rather, aim to "uncover the underlying physical causes for the different summer substructures".

Beginning of section 2.3: At this point I thought your analysis implies some spatial averaging, e.g., a summer season in Switzerland. Only later it becomes clear that this analysis is done grid-point wise. It would help me if you can say this rather early in the text.

Line 134: You could add that D = 92 = the number of days in the summer season.

Line 238: "most regions"? 46% of the NH land area is less than half of the land area, so in what sense is this "most regions"? Did I get something wrong here? The same remark applies to the summary section (line 448).

Line 273: I wonder to what extent this "result" is more or less trivial: To the extent that a particular tercile of the distribution is much more variable than the other two, does this not imply by necessity that an anomalous season must be due to this tercile being anomalous? If this is so (i.e., more or less trivial), you should say this; if I am wrong and this is not trivial, it would help (me, but possibly other readers as well) to explain why it is not trivial. This remark applies equally to the conclusion section (line 407) and the abstract (line 26).

Line 284: "closely"— really? There is quite some resemblance, but I would not call it "close".

Line 364: Is this really a "breaking" trough? In my eyes this is a large (nonlinear) trough, but not quite breaking (yet).

Line 405: Can you speculate why in some areas there is no good correspondence between CESM and ERA-Interim?

**Technical corrections**

Line 302: Should it not read $VF_{cold}$ and $VF_{hot}$?!

Line 588: Is "Earth's Futur." the title of the journal?

———————————————

---

## Referee Comment (RC2) · Anonymous Referee #2 · 9 Jan 2020

Review of "The substructure of extremely hot summers in the Northern Hemisphere"

In this paper, the authors introduce the method of calculating rank day anomalies for each summer in order to characterize the distribution of temperatures during extreme summers. The method, as I understand it, is to sort the 92 daily mean temperature values at each location and then calculate the average at each rank. Then for each summer, the deviation from this climatological mean is taken. They find that in the arctic, extreme summers occur when cold days are warmer than usual and in India, the hottest days drive the anomalously extreme summers. A point that I think is particularly important that is made somewhat in passing is that the characteristics of the extreme summers are consistent with the characteristics of the underlying temperature distributions—there is no obvious regime shift or equivalent for the hottest summers. From this perspective, I think this is a useful tool to verify that we can understand extreme seasons by understanding the underlying temperature distributions.

Overall, I find this study to be worthwhile, but a bit confusing. As the authors state, this is a novel method for looking at extreme summers. They do not spend much time justifying the introduction of such a method, and the advantages it has over examining the local temperature distributions themselves or over methods such as looking at compound heatwaves (Baldwin et al. 2019). Indeed, one of my main takeaway messages from this paper was that extreme summers can be relatively well described by understanding the variance and skewness of the underlying temperature distribution (more below). This method proved that particular point quite nicely. If there are other advantages or conclusions that can be drawn uniquely from these metrics, the authors should highlight them. I believe this paper will be suitable for publication after it addresses the following concerns:

Major:
1) As mentioned above, what is the advantage of this method over more typical examinations of temperature distributions? How does the calculation of RDA differ from quantile analysis? How does the comparison of the contributions of the top 33% and the bottom 33% differ from examining skewness? How does the spatial pattern of XA compare to the spatial pattern of temperature variance? I have included plots based on the ERA-I data I had handy (850 hPa, 1980-2014, 4xdaily), but I think the inclusion of ERA-I surface temperature variance and skewness plots is essential. The comparison with Loikith et al. 2018 is pretty impossible given the size of the panels in their Fig 4.

[Figure]

[Figure]

2) The authors need to better justify not somehow accounting for the trend in summertime temperatures in ERA-I (or better yet, they need to account for the trend). The current justification, i.e., "as we are interested in extreme summers exhibiting the largest absolute T2m anomalies and not the largest T2m anomalies relative to a long-term trend" does not make sense in the context of the later discussion. The analysis as currently presented naturally conflates factors associated with global warming with the dynamics associated with internal modes of climate variability.

> e.g. The point in Nevada has 2016, 2017, and 2018 all included in its five most "extreme" summers. The earliest "extreme" summer there is 2007. Surely, then the signal in RDA is one of global warming. And indeed, if we compare this to the results of McKinnon et al. (2016a) Figure 4, we see a warming of the whole distribution and the largest warming in the bottom quantiles. This then seems to be an examination of the forced response rather than internal variability. Meanwhile the authors argue, quite convincingly, that the extreme summers in India are related to the timing of the monsoon onset, a signal too strong to be dominated by global warming.

3) "Substructure" is not really an appropriate representation of what is studied in this paper. This study is not detailing the relative timing and duration of heatwaves—indeed all temporal ordering is lost in the novel method introduced here. Substructure as I would typically understand it is considered in Fig. 1 and Fig. 8 only. This isn't such a major point about the importance of the paper, but it will require some thought as to a more appropriate term and then significant rewriting.

Other points:

The use of "d" in the equations in combination with the term "substructure" made me mistake "d" for day instead of rank. Consider a different variable name, perhaps? Or explicitly mention that ordering is lost?

l. 144 rewrite for clarity. Perhaps just "allows assessment of"? Consider adding a specific example here.

Consider mentioning which "third" is 30 days so that this calculation is perfectly reproducible
l. 181 Normally → normal

Consider changing the figures so that it is easier to compare ERA-I and CESM. E.g. put Fig 3 a and 4 a together.

Paragraph beginning l. 243: the quantitative spatial correlation value would be helpful here.

Fig. 6: The yellow contours are really difficult to read. Consider having thin dotted lines for continents so that you could use thicker black lines in place of the yellow? Or some other change to make this more readable. Magenta might be better than yellow.

l. 359 normal

Fig. 9: Label lines within the panel b

Analysis of Nevada. Consider work by McKinnon et al. (2016b), which is primarily looking at Eastern US, but their conclusions still seem relevant.

l. 381 Why … the troughs associated with cold anomalies (black contours in 10a) did not occur…

l. 388 It seems like the goal (c.f. Hoskins and Woollings 2015) is to explain the full shape of the temperature PDF, since extreme summers seem consistent with the underlying distribution. But you are correct that a combined approach is necessary for that as well. So maybe just add "… to fully reveal the physical causes of the full shape of the temperature distribution, including extreme summers" or something along those lines?

Paragraph beginning l. 406: This seems like perhaps the major conclusion of this work. Emphasize this more at the beginning.

l. 425 This phrasing is not appropriate. "Often" cannot be determined from these three case studies, and one of the three case studies (US) is in fact a clear case of temperature advection's importance due to an anomalously zonal jet stream.

Paragraph beginning l. 436: This is completely consistent with the eddy advection argument of Garfinkel and Harnik 2017, Tamarin-Brodsky et al. 2019, and Linz et al. 2018

l. 443 New paragraph

l. 445 Not convinced of this (esp. the coherent regions aspect, since mostly this has looked at individual points) by this particular study.

l. 455 A more zonally symmetric/less wavy flow is still a pattern, so this phrasing doesn't really make sense.

References:
Baldwin et al. 2019 Temporally Compound Heat Wave Events and Global Warming: An Emerging Hazard. Doi: 10.1029/2018EF000989

Hoskins and Woollings 2015 Persistent Extratropical Regimes and Climate Extremes. DOI 10.1007/s40641-015-0020-8

McKinnon et al. 2016a, The changing shape of Northern Hemisphere summer temperature distributions doi: 10.1002/2016JD025292

McKinnon et al. 2016b Long-lead predictions of eastern United States hot days from Pacific sea surface temperatures doi:10.1038/ngeo2687

Tamarin-Brodsky et al. 2019 A Dynamical Perspective on Atmospheric Temperature Variability and Its Response to Climate Change. DOI: 10.1175/JCLI-D-18-0462.1

---

## Author Comment (AC1) · 11 Feb 2020

1 **Replies document for reviews of:**

2 **The substructure of extremely hot summers in the Northern Hemisphere**

4 Matthias Röthlisberger[1], Michael Sprenger[1], Emmanouil Flaounas[1], Urs Beyerle[1] and Heini

5 Wernli[1]

6 [1]Institute for Atmospheric and Climate Science, ETH Zürich, Zürich, Switzerland

9 Corresponding author address:

10 Matthias Röthlisberger

11 Institute for Atmospheric and Climate Science,

12 ETH Zürich, Zürich, Switzerland

13 E-mail: matthias.roethlisberger@env.ethz.ch

**General comments to the Reviewers**

We would like to thank both reviewers for their thoughtful and overall encouraging reviews. The reviews were particularly useful for identifying weaknesses in the presentation of the material, but also helped to sharpen our own view of the value of our key results. Major changes that we made to the manuscript include the following: Both reviewers requested the novelties and key insights of this study to be presented more clearly and to account for these comments, we substantially re-worded Section 4. Moreover, we repeated all our analyses after removing a linear trend from all JJA T2m data at each grid point in both data sets, which meant that we also had to redraw all our figures. Note, however, that none of our original conclusions were altered by this detrending. Line numbers mentioned in in this document refer to line numbers in the revised manuscript, unless stated otherwise. Reviewer comments are included below in black font colour and our replies in blue.

**Reviewer 1**

This manuscript investigates Northern Hemisphere extreme hot summer seasons from a statistical point of view. The topic is relevant, because hot summers have societal impact and are going to become more frequent due to anthropogenic climate change. The paper focuses on the entire 3-month summer season rather than addressing individual heat waves (which have been studied before quite extensively). The method involves a novel statistical analysis based on ranking the 92 days of a summer season according to their anomaly with respect to the corresponding climatology. The results indicate that hot summers in different areas on the Northern Hemisphere may have different substructure: in some regions a summer season tends to be hot because the hottest tercile is anomalously hot, while in other regions the summer season tends to be hot because the coldest tercile is anomalously hot. In addition, it is shown that the Community Earth System Model (CESM) is able to broadly represent such regional differences. The regional differences are made plausible by studying a few cases/locations. I think these are interesting results. In addition, the paper is very well written. I have a few specific comments below which may help to produce a final version.

**Comments:**

Major:

1. My only general comment is the following. I found that the statistical method is well described and sounds very interesting, and while reading I was eagerly awaiting the discussion of possible physical causes. But then (reading that section) I was somewhat

disappointed. For instance, the shift in the onset of the Indian monsoon obviously explains the behavior found in the statistical analysis; actually, the explanation is so obvious that in retrospect the statistical analysis almost appears as an artifact. Let me grossly exaggerate to make my point clear: if you have a very simple phenomenon and apply a rather complex or strange analysis to it, you are likely to find a complex or strange result, but the complexity or strangeness of the result in this case would be mostly a feature of the analysis and not a feature of nature. Having said this, I still believe that the analysis is worth doing, and you do it very well.

We agree with the reviewer insofar as in some regions, the physical causes of extreme summers (and their substructure) are very easily understood. However, we do not believe that this jeopardizes the value of the results and novel insights presented in this study. Therefore, we understand this reviewer comment as a call for more clearly highlighting the novel insights derived from this study.

There are four main results of this study that could not have been achieved without developing and applying our novel seasonal anomaly decomposition. First, for each season and grid point, it allows to exactly quantify how much each rank day contributes to the seasonal anomaly or, similarly, how anomalous each rank day was. The key point here is that these results are quantitative and straight forward to understand. For example, our method allows to make statements like: the hottest 30 days of the 2010 summer at the grid point 35°E/58°N were each at least 4 K hotter than their respective rank day mean (i.e., their climatological value, Fig 4e). We expect such local quantitative statements to be particularly relevant for impact studies, as, e.g., excess mortality, ecosystem damages and agricultural yield losses during a particular extreme season conceivably strongly depend on the particular substructure of the extreme season.

Second, our method allows to study the spatial variability in the extreme summer substructure and, furthermore, allows to make statements about the relevance of the coldest, middle and hottest third of extreme summers in a spatially aggregated sense. For example, even though European and US heat waves have been studied widely in the past, it simply has not been known so far that, e.g., in Nevada, the coldest third of the summer days contribute most to extreme summers, while the hottest third of summer days is most important over the UK. Furthermore, it is a novel insight from this study

that almost everywhere in the Northern Hemisphere, the coldest third of the summer contributes substantially (>25%) to extreme summer temperature anomalies. The general relevance of unusually mild summer days for extreme summers is an important result, as it illustrates that we cannot understand extreme summers solely by studying heat waves. Rather, a complete picture of what generates extreme summers must include an understanding of processes operating on longer than synoptic time scales and how they organize different types of synoptic scale-flow features to both prevent cold summer days and foster heat waves.

Third, our study unravels that the mean extreme summer substructure (i.e., averaged over all extreme summers at a particular grid point) can be assessed qualitatively from the variance and skewness of the underlying T2m distribution. This is relevant because there is a large and robust body of literature that has studied the dynamical drivers of the shape of the T2m distribution. Thus, at least qualitatively, the arguments put forward in these studies to explain the T2m distribution shape can also be used to explain the mean extreme summer substructure.

Fourth, we demonstrate that a state-of-the-art climate model (i.e., the CESM1 model) largely reproduces the observed extreme summer substructures. This result testifies to the model's ability to correctly reproduce the dynamical drivers of extreme summers and will be particularly relevant for subsequent studies on extreme summers (and their substructures) in a changing climate.

All of these four points are now made even more explicit in Section 4 (Summary and concluding remarks), in particular on lines 422-436:

[revised manuscript text omitted]

Minor:

1. Line 68: Can you give here an example, too?! You could, for instance, mention Nevada (USA) and say that this will be discussed later.

   We prefer not to give an additional example here for two reasons. First, we call these other possibilities "plausible", as at this stage in the study it is not yet clear whether or not they at all occur. Second, we discuss distinct substructures in much detail on lines 181-231 and would not like to make reference to these examples (which are "results" of this study) already in the introduction.

2. Line 96: Do you really "illustrate physical causes"? I feel that you, rather, aim to "uncover the underlying physical causes for the different summer substructures".

We believe that we indeed "illustrate physical causes", but the second part of the sentence, "in selected regions", is just as important. The phrasing suggested by the reviewer appears to imply that particular extreme summer substructures have particular physical causes, regardless of where on the globe they occur. Given that similar extreme summer substructures can be found e.g., over the northern Sahel region and the high Arctic, we should have stated more clearly that of course distinct physical causes might lead to one and the same extreme summer substructure, provided they occur in different regions.

To account for this comment, lines 474-479 now read: "Clearly, distinct physical causes might lead to similar extreme summer substructures, in particular when comparing regions that are far apart (e.g., the northern Sahel region and the high Arctic, Fig. 5). However, similar extreme summer substructures in neighboring regions conceivably also point to similar physical causes of extreme summers (e.g., the Asian Monsoon region). Therefore, the extreme summer substructure is a helpful tool for discriminating between neighboring regions with distinct physical causes of extreme summers and might also be helpful for identifying coherent regions with similar physical causes of extreme summers."

3. Beginning of section 2.3: At this point I thought your analysis implies some spatial averaging, e.g., a summer season in Switzerland. Only later it becomes clear that this analysis is done grid-point wise. It would help me if you can say this rather early in the text.
   We have added on line 130 "Furthermore, bear in mind that all these quantities are calculated at each grid point individually.".

4. Line 134: You could add that D = 92 = the number of days in the summer season.
   This information is already provided on line 125. We therefore prefer not repeat it on original line 134.

5. Line 238: "most regions"? 46% of the NH land area is less than half of the land area, so in what sense is this "most regions"? Did I get something wrong here? The same remark applies to the summary section (line 448).

The original sentence read: "Overall, Fig. 5c clearly demonstrates that the coldest third of all summer days contributes a substantial fraction to $XA^{ERAI}$ in most regions.". Hence, "most regions" refers to "the regions where the coldest third of all summer days contributes a substantial fraction to $XA^{ERAI}$". The 46% on the other hand, refer to regions where the contribution from the coldest third exceeds the contribution from the hottest third. The question is thus what we are willing to call "a substantial fraction". Figure 5c shows that $XF_{cold}^{ERAI}$ is less than 25% only in very few regions, which is why we stated that almost everywhere it is "substantial".

To clarify this point we have rephrased this sentence (now lines 250-251) to: "Overall, Fig. 5c clearly demonstrates that the coldest third of all summer days contributes a substantial fraction to $XA^{ERAI}$ in most regions [more than 25% over 83% of the Northern Hemisphere land area in ERAI]"

6. Line 273: I wonder to what extent this "result" is more or less trivial: To the extent that a particular tercile of the distribution is much more variable than the other two, does this not imply by necessity that an anomalous season must be due to this tercile being anomalous? If this is so (i.e., more or less trivial), you should say this; if I am wrong and this is not trivial, it would help (me, but possibly other readers as well) to explain why it is not trivial. This remark applies equally to the conclusion section (line 407) and the abstract (line 26).

Indeed, in retrospect, this result is rather easily understood, and thus plausible. However, it is nevertheless certainly relevant, at least for two reasons. First, it is not a priori clear that the climatologically most variable tercile must contribute most to extreme seasons, as also some kind of temperature regime shifts could occur during the most extreme seasons. Such a regime shift can be observed during the 2010 summer at the grid point 35°E/58°N, during which the hottest 30 days exhibited rank day anomalies that were roughly twice as large as during the second most extreme summer (Fig. 3e) and thus clearly showed a different behaviour than in the climatology. The result referred to by the reviewer shows that such regime shifts do not generally occur during extreme summers.

Second, it is not so much the result itself but rather its implications that are non-trivial. The fact that the extreme summer substructure is consistent with the underlying T2m

distribution constrains the possible extreme summer substructures. For example, in a region with strongly negatively skewed temperature distribution, extreme summers are very unlikely to arise from typical "heat waves", but rather must arise from processes that supress cool summer days. However, those processes have hitherto not been studied extensively. We are not aware of any previous study making this point and therefore find it novel and relevant.

We now mention the relevance of this result explicitly on lines 423-436: "Furthermore, a key finding of this study is that the mean extreme summer substructure (i.e., the average substructure of all extreme summers at a particular grid point) is consistent with the shape of the underlying local T2m distribution. The mean extreme summer substructure is largely determined by which of the 92 JJA rank days are most variable (i.e., the rank day variability pattern), which is qualitatively related to the skewness of the T2m distribution. Simply speaking, in regions where the coldest days of the summer are most variable (i.e., negatively skewed T2m distribution), extreme summers occur when the coldest days of the summer are unusually hot, and analogously for the case where hottest days vary the most (i.e., positively skewed T2m distribution). This finding is relevant for two reasons. Firstly, it constrains what kind of extreme summer substructures can locally be expected, in particular in regions with strongly skewed daily temperature distributions. For example, extreme summers arising primarily from extremely hot summer days (i.e., heat waves) are unlikely to occur in regions with strongly negatively skewed temperature distributions. Secondly, some individual extreme summers such as the 2010 summer at the grid point at 35°E/58°N featured clear temperature regime shifts, with rank day anomalies far outside of what could be expected from their climatological variability (e.g., twice as large as the second large anomalies for the same ranks during the 2010 summer at 35°E/58°N). The consistency between the mean extreme summer substructure and the skewness of the (full) T2m distribution illustrates that such regime shifts in the temperature variability during extreme summers are the exception rather than the norm."

7. Line 284: "closely"— really? There is quite some resemblance, but I would not call it "close".

We deleted "closely".

8. Line 364: Is this really a "breaking" trough? In my eyes this is a large (nonlinear) trough, but not quite breaking (yet).

    *Following McIntyre & Palmer (1983) and Martius, Schwierz, & Sprenger (2007) we use "breaking trough" synonymously with "nonlinear trough". It is important to bear in mind that this is a composite trough. Hence, if the composite trough (composited over 100 days) already features meridionally overturning of PV contours, we do feel confident to call it a breaking trough.*

9. Line 405: Can you speculate why in some areas there is no good correspondence between CESM and ERA-Interim?

    *The reviewer raises a very interesting question, which in our opinion might warrant a subsequent study. However, as the reviewer points out quite rightly, based on the presented results we could only speculate about why CESM and ERAI extreme summer substructures disagree in some regions. We are concerned that speculation about this point might lead to more confusion than clarity and therefore refrain from doing so.*

10. Line 302: Should it not read V Fcold and V Fhot?!

    *Yes, indeed, many thanks for spotting this error! We have changed this according to the reviewer comment.*

11. Line 588: Is "Earth's Futur." the title of the journal?

    *Yes, the title of the journal is "Earth's Future", which is abbreviated in the WCD citation style to "Earth's Futur.". The paper can be accessed under: https://agupubs.onlinelibrary.wiley.com/doi/full/10.1029/2019EF001189*

**Reviewer 2**

Summary: In this paper, the authors introduce the method of calculating rank day anomalies for each summer in order to characterize the distribution of temperatures during extreme summers. The method, as I understand it, is to sort the 92 daily mean temperature values at each location and then calculate the average at each rank. Then for each summer, the deviation from this climatological mean is taken. They find that in the arctic, extreme summers occur when cold days are warmer than usual and in India, the hottest days drive the anomalously extreme

summers. A point that I think is particularly important that is made somewhat in passing is that the characteristics of the extreme summers are consistent with the characteristics of the underlying temperature distributions—there is no obvious regime shift or equivalent for the hottest summers. From this perspective, I think this is a useful tool to verify that we can understand extreme seasons by understanding the underlying temperature distributions. Overall, I find this study to be worthwhile, but a bit confusing. As the authors state, this is a novel method for looking at extreme summers. They do not spend much time justifying the introduction of such a method, and the advantages it has over examining the local temperature distributions themselves or over methods such as looking at compound heatwaves (Baldwin et al. 2019). Indeed, one of my main takeaway messages from this paper was that extreme summers can be relatively well described by understanding the variance and skewness of the underlying temperature distribution (more below). This method proved that particular point quite nicely. If there are other advantages or conclusions that can be drawn uniquely from these metrics, the authors should highlight them. I believe this paper will be suitable for publication after it addresses the following concerns:

**Major:**

1. As mentioned above, what is the advantage of this method over more typical examinations of temperature distributions? How does the calculation of RDA differ from quantile analysis? How does the comparison of the contributions of the top 33% and the bottom 33% differ from examining skewness? How does the spatial pattern of XA compare to the spatial pattern of temperature variance? I have included plots based on the ERA-I data I had handy (850 hPa, 1980-2014, 4xdaily), but I think the inclusion of ERA-I surface temperature variance and skewness plots is essential. The comparison with Loikith et al. 2018 is pretty impossible given the size of the panels in their Fig 4.

   There are several ways in which our method differs from the standard characterizations of the T2m distribution listed by the reviewer and which make our method a valuable tool that is complementary to standard methods.

   A first difference lies in the purpose of the method we developed. The novelty of this study is that it assesses how entire summer seasons become extreme from a statistical (and partly dynamical) point of view. This research question is certainly relevant as recent extreme summers had large societal impacts (going beyond the impacts of

individual heat waves) and which therefore call for a better understanding of extreme summer seasons overall. In the process of addressing our research question we learned that the mean extreme summer substructure at a particular grid point can be inferred qualitatively from the skewness of the underlying daily temperature distribution. As the reviewer quite rightly noticed, this is a very important result of this study which could not have been anticipated beforehand. However, for any study, the choice of method is driven by the purpose of the study and not by its final results. Therefore, the quantities and methods we work with ($RDA, XA,$ etc.) are natural and meaningful choices for addressing our research question, and their development and application was imperative for arriving at the understanding of extreme summers that we now have.

Second, our method does not only allow to analyse the mean behaviour of extreme seasons at a particular grid point (i.e., averaged over all extreme seasons at a particular grid point) but can also characterize individual extreme seasons. Figures 4b,c show two examples of distinct extreme summer substructures occurring at one particular grid point. In such regions, the ability to characterize individual extreme seasons is certainly an advantage over simply characterizing the mean extreme season. Furthermore, the degree to which different extreme summers at a particular grid point resemble each other cannot be inferred from considering skewness and variance of a particular T2m distribution, but this information is readily available after employing the method developed here. For this particular purpose, quantile analysis would certainly be a valid alternative (which we actually tested in an earlier stage of this work). However, the method developed in this study allows for an exact decomposition of the seasonal anomalies, which does not rely on any quantile function and which we therefore consider to be more elegant.

Third, we believe that the quantitative results of our seasonal anomaly decomposition are particularly straightforward to understand. For example, Fig. 4e reveals that at 35°E/58°N the hottest 30 days of the 2010 summer were each at least 4 K warmer than the climatological values of the 30 hottest days, which for some ranks is more than 2 K more than the second hottest summer. Moreover, at the grid point in India, the hottest third of the summer 2005 contributed 95% of the seasonal mean anomaly. Finally, we show that over 46% of the land mass, the coldest third of extreme summers contributes more to the extreme summer anomaly than the hottest third. We do not see how such

exact quantitative statements on the substructure of extreme summers could be achieved based on the analyses suggested by the reviewer. However, we strongly believe that such exact quantitative statements are valuable and convey particular characteristics of extreme summers in a very intuitive way.

In order to account for this comment, we have substantially reworded and extended the summary and concluding remarks (Section 4, lines 403-509) and moreover included Fig. S1 which shows the skewness of the daily T2m distribution in ERAI (Fig. S1 is also included at the end of this document). Furthermore, we now clearly state whether we discuss mean extreme summer substructures (i.e., averaged over all extreme summers at a particular grid point) or the substructure of a particular summer.

2. The authors need to better justify not somehow accounting for the trend in summertime temperatures in ERA-I (or better yet, they need to account for the trend). The current justification, i.e., "as we are interested in extreme summers exhibiting the largest absolute T2m anomalies and not the largest T2m anomalies relative to a longterm trend" does not make sense in the context of the later discussion. The analysis as currently presented naturally conflates factors associated with global warming with the dynamics associated with internal modes of climate variability. e.g. The point in Nevada has 2016, 2017, and 2018 all included in its five most "extreme" summers. The earliest "extreme" summer there is 2007. Surely, then the signal in RDA is one of global warming. And indeed, if we compare this to the results of McKinnon et al. (2016a) Figure 4, we see a warming of the whole distribution and the largest warming in the bottom quantiles. This then seems to be an examination of the forced response rather than internal variability. Meanwhile the authors argue, quite convincingly, that the extreme summers in India are related to the timing of the monsoon onset, a signal too strong to be dominated by global warming.

We agree with the reviewer on this point. The intention of this study was to understand how the most extreme (i.e., often recent) summers became so extreme. However, the reviewer is right, not detrending the T2m data might obscure the causes of extreme summers arising from internal variability.

To account for this comment, all analyses have been repeated after removing a linear trend from JJA data, separately at each grid point and in both data sets. The new Figs.

2, 8 and 9 have been produced with non-detrended data, as for these figures absolute values of T2m are either more intuitively understood (Figs. 2 and 8) or the absolute value of T2m is relevant (Fig. 9). However, also for these Figures, extreme seasons have been identified based on the detrended data.

None of our conclusions are altered by this detrending. However, some of the archetypical extreme seasons that we used to illustrate archetypical extreme seasons in the original Section 3.2 no longer appear as extreme seasons. Therefore, in Figs. 3d,e and 4d,e we now show results for the grid points closest to Paris (2°E/49°N) and at 35°E/58°N). Note further that for the Nevada grid point the rank day variability pattern remains almost unchanged, even though the extreme seasons are now more evenly spread throughout the ERAI period.

3. "Substructure" is not really an appropriate representation of what is studied in this paper. This study is not detailing the relative timing and duration of heatwaves—indeed all temporal ordering is lost in the novel method introduced here. Substructure as I would typically understand it is considered in Fig. 1 and Fig. 8 only. This isn't such a major point about the importance of the paper, but it will require some thought as to a more appropriate term and then significant rewriting.

We disagree with the reviewer here but nevertheless appreciate this comment as it points to a possible source of confusion that we wish to avoid in the revised manuscript. To our knowledge, the term "season substructure" is not (yet) a widely used term in atmospheric sciences. In particular, "season substructure" does not necessarily need to have some kind of temporal meaning. Therefore, we allow ourselves to use this term in a way that does not relate to temporal ordering but that we nevertheless do find appropriate and meaningful.

Arguably, the term "season substructure" implies some kind of disaggregation of a season into its sub-parts. Consequently, studying the "season substructure" means studying particular aspects of these sub-parts. Admittedly, one such disaggregation could be temporal and in this case, studying the season substructure would indeed mean studying, e.g., the early, middle and late parts of the season (or any other consecutive time periods during the season, such as individual heat waves). However, equally well this disaggregation could be with regard to temperature or any other variable. In this

case, studying the substructure of a season means studying particular aspects of the cold, middle and warm parts of this season. This is exactly how we use this term in this study and we therefore think that it indeed is appropriate.

However, we have realized that in Fig. 1 and on original lines 60-69 we unintentionally implied a disaggregation over time, which of course was misleading the reader. We have therefore adjusted the schematic in Fig. 1 and rewritten the original lines 60-69 (now lines 60-69) to: "Like any other summer, an extreme summer will inevitably contain cooler and hotter days, which constitute the upper and lower parts of the T2m distribution during that summer. However, it is currently not known which part of the T2m distribution is particularly anomalous during an extreme summer. Thus, extreme summers with distinct "substructures" might occur, some of which are schematically illustrated in Fig. 1. For example, a summer might be an extreme summer because the hottest days of the season are particularly anomalous, with the remainder of the summer days being only moderately warmer than or even close to climatology. Such an extreme summer substructure was observed in large parts of Europe in the summer 2015, when the anomalies of the seasonal hottest days exceeded those of the seasonal mean by almost a factor of two (Dong et al., 2016). Hence, the hottest days of the 2015 summer contributed over proportionally to the seasonal mean anomaly. However, also other substructures are plausible: a suppression of cool summer days, a uniform shift in the entire summer temperature distribution or any combination of these three options.".

Other points:

1. The use of "d" in the equations in combination with the term "substructure" made me mistake "d" for day instead of rank. Consider a different variable name, perhaps? Or explicitly mention that ordering is lost?

   We have rephrased the sentence on line 134, which now reads: "…T2m value with rank d in season k (i.e., the temporal ordering of the days is lost, see Fig. 2b)."

2. 144 rewrite for clarity. Perhaps just "allows assessment of"? Consider adding a specific example here.

   An example has been added on lines 146-148: "For example, if for a particular season $k$ $SA_k = 1$ K and $RDA_{92,k} = 3$ K (i.e., the hottest day of season $k$ is 3 K warmer than

the respective rank day mean) this day contributed $3/92 = 0.0326$ K or 3.26% to the seasonal anomaly $SA_k$."

3. Consider mentioning which "third" is 30 days so that this calculation is perfectly reproducible

Lines XY read: "The notation $[x]$ hereby stands for $x$ rounded to the nearest integer. For computing contributions to $SA_k$ from the middle and hottest thirds of the summer days ($SF_{middle,k}$ and $SF_{hot,k}$), the sum in Eq. (5) runs from $\left[\frac{D}{3}\right] + 1$ to $\left[D\frac{2}{3}\right]$ for $SF_{middle,k}$ and from $\left[D\frac{2}{3}\right] + 1$ to $D$ for $SF_{hot,k}$." From this statement it is clear that the coldest "third" only contains 30 days.

4. 181 Normally -> normal

Changed as requested by the reviewer.

5. Consider changing the figures so that it is easier to compare ERA-I and CESM. E.g. put Fig 3a and 4 a together.

We have considered this option but we find it more intuitive to present the anomalies $XA^M$ alongside with their respective decompositions and therefore chose to stick to the original figure layout for Figs. 3 and 4.

6. Paragraph beginning l. 243: the quantitative spatial correlation value would be helpful here.

We are not entirely sure what measure of spatial correlation the reviewer has in mind exactly. We prefer to leave this passage as it was originally.

7. Fig. 6: The yellow contours are really difficult to read. Consider having thin dotted lines for continents so that you could use thicker black lines in place of the yellow? Or some other change to make this more readable. Magenta might be better than yellow.

We have changed the contour color to green and adjusted the contour levels to make them more readable.

8. l. 359 normal

Changed as requested.

9. Fig. 9: Label lines within the panel b

Labels have been added in panel b.

10. Analysis of Nevada. Consider work by McKinnon et al. (2016b), which is primarily looking at Eastern US, but their conclusions still seem relevant.

The work of McKinnon et al. (2016b) is certainly most interesting, in particular if one aimed at predicting the substructure of summer with a seasonal forecasting system. For the Nevada grid point, however, we do not see how exactly the work of McKinnon et al. (2016b) relates to our analysis, since their focus is primarily on the Eastern US and on predicting hot days from a particular tropical SST pattern.

11. l. 381 Why … the troughs associated with cold anomalies (black contours in 10a) did not occur…

The "right phasing" seems crucial to us here and there might well have been troughs with associated cold anomalies during these extreme summers, just not over the Nevada region. Therefore, we prefer our original formulation here.

12. l. 388 It seems like the goal (c.f. Hoskins and Woollings 2015) is to explain the full shape of the temperature PDF, since extreme summers seem consistent with the underlying distribution. But you are correct that a combined approach is necessary for that as well. So maybe just add "… to fully reveal the physical causes of the full shape of the temperature distribution, including extreme summers" or something along those lines?

The reviewer comment is correct insofar as "fully revealing the physically causes of the full temperature distribution" would also help to understand the physical causes of extreme summers. This paper, however, focusses first and foremost on extreme summers and therefore we prefer to stick to our original wording.

13. Paragraph beginning l. 406: This seems like perhaps the major conclusion of this work. Emphasize this more at the beginning.

We now emphasize this result much more in Section 4.

14. l. 425 This phrasing is not appropriate. "Often" cannot be determined from these three case studies, and one of the three case studies (US) is in fact a clear case of temperature advection's importance due to an anomalously zonal jet stream.

We have changed the wording to (now line 450-452) "However, three case studies illustrate that the extreme summer substructure cannot always be explained by temperature advection alone."

15. Paragraph beginning l. 436: This is completely consistent with the eddy advection argument of Garfinkel and Harnik 2017, Tamarin-Brodsky et al. 2019, and Linz et al. 2018.

We agree. Reference is now made to all three studies on lines 468-469: "This result is consistent with previous work on physical causes of non-Gaussian temperature distributions (Garfinkel and Harnik, 2017; Linz et al., 2018; Tamarin-Brodsky et al., 2019), as it highlights the role of temperature advection by transient waves in generating a non-uniform rank day variability pattern, or similarly, a skewed T2m distribution."

16. l. 443 New paragraph

Changed as requested.

17. l. 445 Not convinced of this (esp. the coherent regions aspect, since mostly this has looked at individual points) by this particular study.

We have rephrased this paragraph in order to be more precise about how the extreme summer substructure might help to delineate coherent regions with similar drivers of extreme summers.

Lines 474-479 now read: "Clearly, distinct physical causes might lead to similar extreme summer substructures, in particular when comparing regions that are far apart (e.g., the northern Sahel region and the high Arctic, Fig. 5). However, similar extreme summer substructures in neighboring regions conceivably also point to similar physical causes of extreme summers (e.g., the Asian Monsoon region). Therefore, the extreme summer substructure is a helpful tool for discriminating between neighboring regions with distinct physical causes of extreme summers and might also be helpful for identifying coherent regions with similar physical causes of extreme summers."

18. l. 455 A more zonally symmetric/less wavy flow is still a pattern, so this phrasing doesn't really make sense.

It is true that a less wavy flow is still a flow pattern but it is not "the particular flow pattern necessary to produce cold summer days". The point here really is to say that for producing abnormally mild coldest summer days no special flow pattern is required. Rather, the particular pattern that usually generates the coldest summer days just does not occur.

To clarify this point we have rephrased lines 488-491 to: "However, as illustrated by the example of extreme summers in the western US, the processes that suppress the occurrence of cold summer days sometimes seem rather intangible, as they do not necessarily manifest themselves in the occurrence of an unusual flow pattern, but rather in the non-occurrence of the particular flow that typically produces the coldest summer days.".

Martius, O., Schwierz, C., & Sprenger, M. (2007). Dynamical tropopause variability and potential vorticity streamers in the Northern Hemisphere — A climatological analysis. *Adv. Atmos. Sci.*, *25*(3), 367–380. https://doi.org/10.1007/s00376-008-0367-z

McIntyre, M. E., & Palmer, T. N. (1983). Breaking planetary waves in the stratosphere. *Nature*, *305*(5935), 593–600. https://doi.org/10.1038/305593a0

[Figure]

**Figure S1.** Skewness of daily T2m in ERA-Interim. Black crosses as in Fig. 3a.